# Towards practical AI for agriculture: A self-supervised attention framework for Spinach leaf disease detection

Nilavro Das Kabya ⓘ, MD Shaifullah Sharafat ⓘ, Rahimul Islam Emu ⓘ, Mehrab Karim Opee, Riasat Khan ⓘ*

Electrical and Computer Engineering, North South University, Dhaka, Bangladesh

* riasat.khan@northsouth.edu

## Abstract

Malabar spinach is a nutrient-dense leafy vegetable widely cultivated and consumed in Bangladesh. Its productivity is often compromised by Alternaria leaf spot and straw mite infestations. This work proposes an efficient and interpretable deep learning framework for automatic Malabar spinach leaf disease classification. A curated dataset of Malabar spinach images collected from Habiganj Agricultural University and supplemented with public samples was categorized into three classes: Alternaria, straw mite, and healthy leaves. A lightweight SpinachCNN established a strong baseline, while Spinach-ResSENet, enhanced with squeeze-and-excitation modules, improved channel-wise attention and feature discrimination. A customized Vision Transformer (SpinachViT) and SwinV2-Base were further investigated to assess the benefits of transformer-based architectures under limited data. To mitigate annotation scarcity, we employed SimSiam-based self-supervised pretraining on unlabeled images, followed by supervised fine-tuning with cross-entropy or a hybrid objective combining cross-entropy and supervised contrastive loss. The best-performing domain-optimized model, SimSiam-CBAM-ResNet-50, incorporated Convolutional Block Attention Modules and achieved 97.31% test accuracy, 0.9983 macro ROC-AUC, and low calibration error, while maintaining robustness to Gaussian and salt-and-pepper noise. Although a SwinV2-Base benchmark pretrained on ImageNet-22k reached slightly higher accuracy (97.98%, 98.99% with test-time augmentation), its 86.9M parameters and reliance on large-scale pretraining reduce feasibility for edge deployment. In contrast, the SimSiam-CBAM model offers a more parameter-efficient and deployment-friendly solution for real-world agricultural applications. Model decisions are interpretable via Grad-CAM, Grad-CAM++, and Layer-CAM, which consistently highlight biologically relevant lesion regions. The spinach dataset used in this study is publicly available on: https://huggingface.co/datasets/saifullah03/malabar_spinach_leaf_disease_dataset.

**Data availability statement:** The spinach dataset used in this study is publicly available on: https://huggingface.co/datasets/saifullah03/malabar_spinach_leaf_disease_dataset.

**Funding:** The author(s) received no specific funding for this work.

**Competing interests:** The authors have declared that no competing interests exist.

# 1 Introduction

Malabar spinach is a significant vegetable in Bangladesh, valued for its economic and nutritional benefits [1]. However, its growth is hindered by substantial yield losses due to Alternaria leaf spot and straw mite infestations [2]. While computer vision techniques have improved disease diagnosis in key staple crops [3], automated solutions for Malabar spinach have not been well explored. Farmers depend on manual inspections and late interventions. Prior studies on Bangladeshi agriculture have primarily focused on rice, wheat, and common horticultural species. Systematic datasets and models for Malabar spinach are largely absent. This absence limits academic understanding of the crop's pathology and the practical deployment of digital decision-support tools in local farming communities.

Alternaria is considered the most detrimental pathogen, and it causes significant threats to essential crops, including tomato, potato, cabbage, carrot, wheat, apple, peanut, cotton, chili pepper, grapes, sunflower, onion, and lettuce [4]. In Bangladesh, Alternaria diseases significantly threaten crop productivity and quality, especially during the humid monsoon season when fungal growth is favored [5]. Traditional disease detection methods are slow, labor-intensive, and prone to errors in large-scale farming. Using image augmentations, deep learning techniques, and recent computer vision detection [6] can have promising potential in the early detection and identification of diseases [7].

This research explores the latest computer vision approaches for detecting Alternaria diseases in crops commonly affected in Bangladesh. An automated, cost-effective solution, that makes use of image recognition with classification and deep learning techniques, is proposed to help farmers detect diseases and take necessary actions [8]. Straw mites are among the most significant risks to crop health, contributing significantly to reduced quality and yield. Straw mite infestations are typically accompanied by fine webbing and tiny, nearly invisible mites. Cereals are mainly affected by these pests, causing the discoloration, shriveling and necrosis of plant tissues. Such often leads to stunted growth and lower yields. Identification and distinction between damages caused by straw mites, Alternaria disease, and healthy crops would help to give precision in agricultural interventions [9].

In the case of Alternaria, early symptoms include the appearance of tiny black, pin-sized specks on the leaves and stems. The circular lesions are brown with concentric rings and a dark necrotic center surrounded by a yellow halo, which serves as a key visual marker for detection. These spots gradually enlarge and coalesce, spreading from older, lower leaves upward, ultimately resulting in irregularly shaped necrotic areas. For straw mite infestation, early signs include webbing on the leaf surfaces and a characteristic silver-white discoloration caused by chlorophyll depletion. The present study applies advanced deep learning methods to classify spinach leaves into three categories: healthy, infected with Alternaria, and infected with straw mite. Multiple image datasets and state-of-the-art neural networks are utilized to develop a more generalized and scalable solution. The proposed approach is intended to support early detection and enable farmers to implement timely and effective disease management strategies, thereby safeguarding crop health and productivity.

This work proposes Malabar spinach disease classification using efficient deep learning techniques on a dataset of spinach leaves affected by Alternaria and straw mite. The following are the key contributions of this work:

- A merged raw dataset comprising slightly over 700 Malabar spinach leaf images was constructed by integrating a private dataset from the Plant Pathology Department of Habiganj Agricultural University with a publicly available dataset; after data augmentation, this expanded to more than 2,100 effective training samples.
- A unique approach of Spinach Res-SENet architecture leverages the power of residual connections and squeeze-and-excitation modules, immensely improving feature representation and classification accuracy.
- The cross entropy loss is replaced with a hybrid training loss that combines Supervised Contrastive Loss and Label-Smoothed Cross-Entropy for better generalization, and it demonstrated improved class separation, which leads to robustness over standard loss formulations
- To further improve model robustness and calibration under real-world conditions, synthetic noise, such as Gaussian and Salt-and-Pepper noise, is injected during training as part of an ablation study.

The novelty of this work is to implement a domain-optimized pipeline for Malabar spinach disease classification by integrating a merged dataset, self-supervised SimSiam, a custom Res-SENet architecture, and a hybrid loss function with noise augmentation techniques.

This article has been divided into the following sections. Sect 2 discusses the notable research work done in the field of leaf disease detection using various machine learning and deep learning techniques. Sect 3 provides a detailed description of this work's methodology, including data collection, preprocessing, the use of various models and how they were modified according to the dataset. Sect 4 shows a comparative analysis of the models used, the results obtained, and what they portray about the model.

## 2 Literature review

Despite advances in computer vision for diagnosing diseases in major staple crops, Malabar spinach has received little attention in terms of automated solutions. Existing research in the context of Bangladeshi agriculture has predominantly focused on rice, wheat, and widely cultivated horticultural species, with limited systematic datasets or deep learning models available for Malabar spinach. This section reviews recent studies on deep learning-based approaches to automatic crop disease detection.

### 2.1 Deep learning in major food crops

Joseph et al. [10] implemented a deep learning-based plant disease classification approach for rice, wheat, and maize crops. Their results showed that data augmentation effectively reduced overfitting, achieving classification accuracies of 97% for rice, 98% for wheat, and nearly 99% for maize. Among the tested models, Xception and MobileNet achieved 95.8% and 94.6% accuracy, respectively, for maize diseases, while wheat disease classification reached 96.3%. Rice disease classification yielded the highest performance, with Xception achieving 97.2% and InceptionV3 achieving 96.2%. Girmaw et al. [11] applied transfer learning for plant disease detection, initially using a dataset of 1,400 images, which was later expanded to 3,500 to enhance model performance and reduce overfitting. EfficientNetB7 emerged as the top-performing model due to its compound scaling and advanced architectural efficiency.

### 2.2 Lightweight and real-time plant disease detection

Bhagat et al. [12] applied a lightweight deep learning model for real-time detection of diseases in the pigeon pea, which is optimized for resource-constrained devices. Using the first real-field pigeon pea dataset with 12,232 images across five disease classes, Lite-MDC employed MDsConv for multiscale feature extraction while minimizing computational complexity. With only 2.2 million parameters, it achieved 94.14% accuracy, outperforming ResNet-50 and DenseNet121.

 

The model was tested on publicly available datasets such as Plant Village and Cassava. The lightweight model made it very feasible for deployment on drones and smartphones, making it suitable for real-time use. Obsie et al. [13] designed an enhanced model for mummy berry disease detection in wild blueberries, incorporating the Coordinate Attention mechanism into the base model for such challenges as occlusion, complex backgrounds, and limited data from field collections. It reached a better precision and recall of 96.3% on synthesized augmented images compared to its baseline model, Yolov5s, which detects infected plant parts at varied spatial scales. The CA module strengthened the discrimination of features, while the CIoU loss enhanced the bounding box regression and increased the robustness in field environments. Lightweight Yolov5s-CA ensured real-time applicability, worked effectively with combined synthetic and real datasets, reduced labor-intensive data collection, and has excellent potential in wide deployment over resource-constrained devices and broader plant disease management applications.

### 2.3 Deep learning applications in leaf disease classification

Nikitha and his colleague [14] developed a soybean leaf disease identification system. Using image segmentation and K-means clustering, the CNN model attained the highest accuracy of 96%, while SVM and KNN achieved 76% and 64%, respectively. The study showed CNN's superior performance due to its ability to extract features and effectively handle the multi-class classification challenge. Chowdhury et al. [15] presented the detection of plant leaf disease with the help of deep learning techniques in agriculture in Bangladesh. The authors used the Plant Village dataset, which consisted of 17,430 images from bell peppers, tomatoes, and potato leaves. The applied custom CNN model performed well in crop disease classification, achieving 85.31% accuracy, outperforming the other models. The authors developed a mobile application for real-time disease detection to aid Bangladeshi farmers.

Nigar et al. [16] used the EfficientNetB0 deep learning model for the classification of 38 plant diseases. The system achieved the highest accuracy, precision, and recall rates of 99.69%, 98.27%, and 98.26%, respectively. LIME explainable AI was used to provide visual insights into the classifier's decisions and provide more insight and interpretability. EfficientNetB0 outperformed other models such as MobileNetV2, CNN, and ResNet-50. Finally, they developed a mobile application called "PlantCare" for real-time disease detection, making this approach practical for farmers. Madhurya and Jubilson [17] presented a more effective deep learning framework, YR2S, for the detection and classification of plant leaf diseases with an accuracy of 99.69%. Optimized by Enhanced Rat Swarm Optimizer (ERSO), the YR2S framework included YOLOv7 for identifying diseases with the ShuffleNetV2 classifier. It combined PCFAN for feature extraction, which captured color, shape, size, edge, and morphological features, and uses the Red Fox Optimization Algorithm to segment diseased areas accurately. The proposed framework outperformed existing frameworks, including DenseNet121 and hybrid VGG16-ResNet-50, in terms of both accuracy and computational efficiency when tested on the New Plant Diseases Dataset comprising 87,867 images over 38 classes.

Salam et al. [18] applied deep learning methods for formulating diseases in mulberry leaves, especially for healthy leaves, leaf rust, and leaf spots. The authors enhanced feature extraction and reduced overfitting through transfer learning applied to a dataset containing 1091 images, augmented to 6000 images. The modified MobileNetV3Small attained the highest accuracy of 97.0% among all other models. Xiao et al. [19] proposed the SE-VRNet, a lightweight deep residual network agile in attention mechanisms to support feature extraction for leaf disease recognition. The model achieved top-1 classification accuracies of 99.73% and 95.71% on NewData and SelfData datasets, respectively. The authors enhanced model generalization and robustness using a dataset expansion strategy with increased image sample numbers to 87867. SE-VRNet amplification outperformed other classical architectures like ResNet, VGG16, and GoogLeNet due to the more significant yet compact model size appropriate for mobile deployment. Such findings captured the essence of AI for mobile agriculture based on the understanding of deep learning, lightweight and deep learning efficient models. Salam et al. [20] implemented the MobileNetV3 deep learning algorithm to predict the leaf diseases of mulberry with an accuracy of 94.2% and 96.4% for the leaf rust and leaf spot classification, respectively. Transfer learning with a training set of 6,000 images increased feature extraction processes and reduced overfitting.

From the literature review, it can be concluded that deep learning techniques have not been used for Malabar spinach leaf disease classification. Significant works have been done on staple crops and region-specific applications utilizing efficient architectures, interpretable frameworks and smartphone-based automatic disease detection. This highlights a clear research gap and the need for a domain-optimized, interpretable, and lightweight deep learning framework for underrepresented crops.

## 3 Methodology

### 3.1 Dataset description

The dataset used for this study was the Malabar spinach leaf dataset, consisting of three categories: healthy, infected by Alternaria, and infected by straw mites. This dataset was established by combining Habiganj Agricultural University's private collection with a public straw mite dataset obtained from Daffodil International University. The raw corpus contained slightly more than 700 unique leaf images; after applying the augmentation pipeline described in Sect 3, the effective training set contained over 2,100 samples. The entire combined set was split once into training, validation, and test sets at a 70–15–15 ratio. Exactly the same partition was then used throughout all subsequent experiments, including self-supervised and transformer-based models. Each split was prepared in three class-specific folders according to the standard ImageFolder arrangement. Model pipelines like SimSiam or SwinV2-Base acted on this very same 70–15–15 split, their variations solely coming from their respective internal pretraining or augmentation, not from any alternative split of the dataset. Sample images of the collected spinach leaf disease dataset are illustrated in Fig 1.

### 3.2 Data preprocessing and augmentation

Data preprocessing is necessary for enhancing image features and improving model performance by removing unwanted objects and standardizing dataset quality. Fig 2 illustrates sample training images after applying the image augmentation techniques. Images were resized to 224×224 pixels, with a focus on cropping diseased leaf areas. Increased model robustness is achieved by applying different augmentation techniques only on the training images, such as horizontal flipping, rotation, brightness adjustment, contrast enhancement, adding Gaussian noise, and perspective transformations. Enhancement operations are used to increase clarity and decrease noise, which includes sharpening and Gaussian blur. Normalization of pixel values was set to a range between 0 and 1, and color space manipulation, such as grayscale and HSV, was opted for to enhance the discriminative features. It mainly uses Python and ImageJ for data cleaning and reliability; therefore, the class imbalance problem could be dealt with through techniques such as oversampling or undersampling. Fig 3 illustrates the class distribution of the Malabar spinach leaf dataset in percentage, showing that

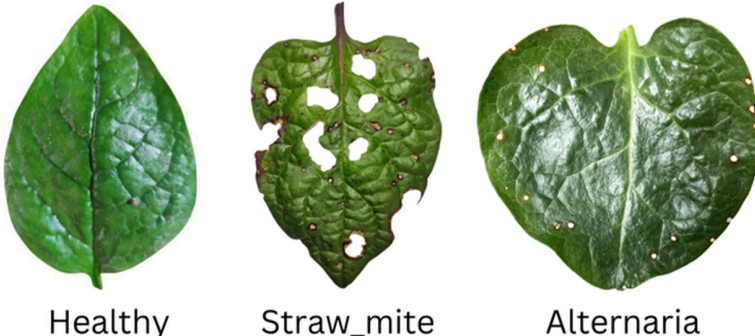

**Fig 1**. **Sample images of the collected spinach dataset.**

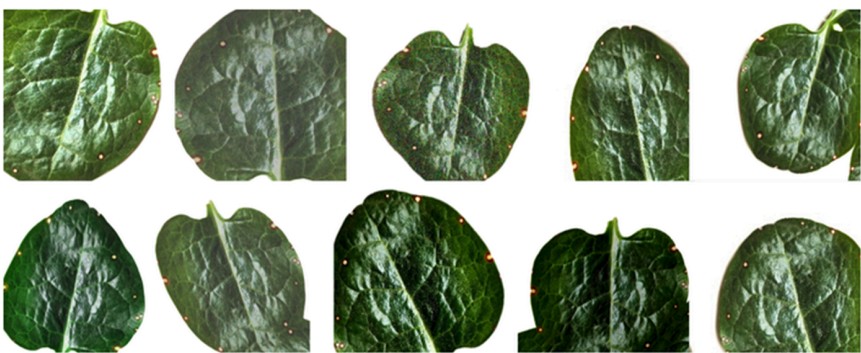

**Fig 2**. Augmented training images of the employed dataset.

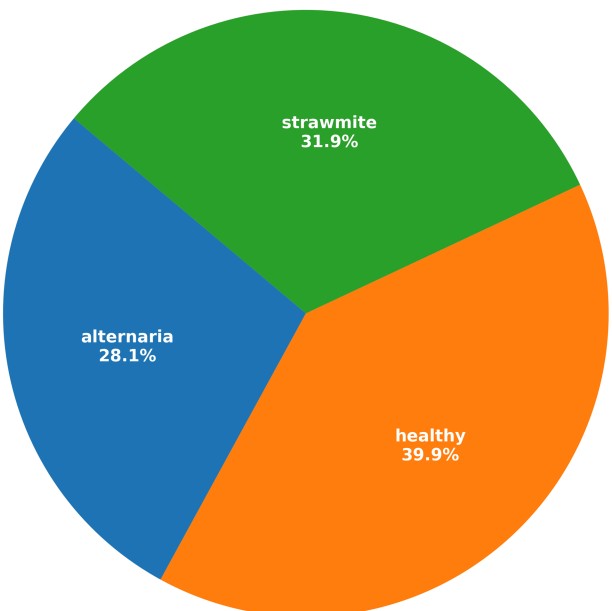

**Fig 3**. Percentage-wise class distributions of the employed dataset.

healthy samples constitute the largest portion (39.9%), followed by straw mite (31.9%) and Alternaria (28.1%), indicating a moderately imbalanced dataset. Test-time augmentation is applied only to the test split at inference time, and no augmented view of any test image is ever used during training.

### 3.3 Applied deep learning models

Various state-of-the-art pretrained deep learning models, i.e., MobileNetV2, EfficientNetB0, ResNet-50, and ViT-B/16, were used for feature extraction and additional fine-tuning in this research. EfficientNetB0 [21] employs a compound scaling approach, which effectively balances accuracy and computational cost. ResNet-50 [22], a deep residual network, employs skip connections to facilitate enhanced feature learning. ViT-B/16 [23], a transformer-based model, takes images in patch format and employs self-attention mechanisms. All models were initialized with ImageNet weights, their final

classifier layer was replaced with a three-class softmax classifier, and they were fine-tuned with appropriate optimizers and learning rates.

**3.3.1 Spinach CNN.** The custom SpinachCNN model, illustrated in Fig 4, comprises an initial convolutional layer of 7×7 with a stride of 2, Batch Normalization, ReLU, and Max Pooling. The network is then followed by three residual blocks, each increasing the feature channels from 64 to 128, 128 to 256, and 256 to 512, respectively. A global pooling layer was equipped with average and max pooling for feature extraction. Two fully connected layers follow this step: batch normalization, ReLU activation, and dropout. Finally, a softmax-activated output layer classifies the input into three categories.

The SpinachCNN model was designed and trained using an optimal and reliable experimental setup. For the process of optimization, AdamW optimizer was utilized, where the base learning rate was $1 \times 10^{-4}$, and the weight decay value was $1 \times 10^{-4}$, and it had a batch size of 32 for 30 epochs. In the present code, no learning rate scheduler and early stopping were applied, which trained the code from start to finish for the total number of epochs. For the process of optimization, Cross-Entropy Loss was employed, which utilized label smoothing of 0.1, and mixed precision training with AMP gradient scaling. In the data pipe, extensive augmentation was performed during the process, which involved randomized converted crops, flipped images, rotated images, jitter, and affine transforms, where the images were converted based on the standard mean and standard deviation of the dataset. In the code for an optimal and reliable experimental setup, the random seed was set to 42 for the entire Python, NumPy, and Torch.

**3.3.2 Spinach Res-SENet model.** Spinach Res-SENet model, shown in Fig 5, classifies leaf images efficiently using residual connections and squeeze-and-excitation (SE) blocks [24], which makes it possible to represent the features robustly. This architecture consists of an input layer that accepts three-channel images of 224×224, which are preprocessed before entering this network. The initial stage of this model comprises the Convolution Block, which performs a 2D convolution with a kernel size of 7×7 and 64 output channels. Next, batch normalization and a ReLU activation function are applied for non-linearity. A max-pooling layer downsamples the spatial size to obtain only coarse-level features while reducing the computational cost.

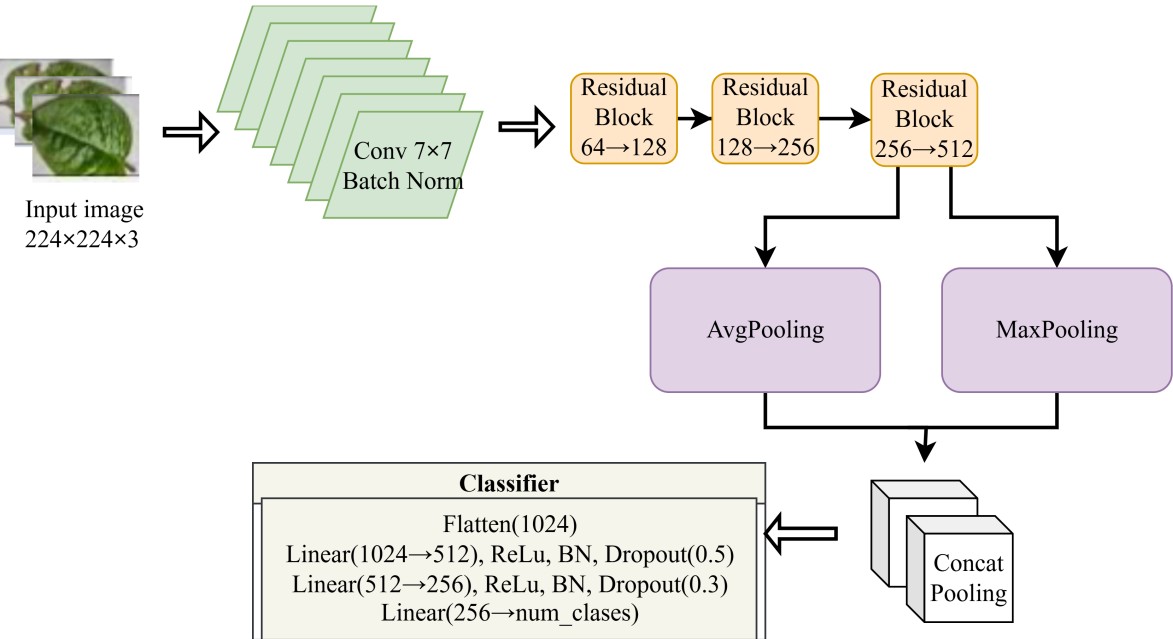

**Fig 4**. Proposed SpinachCNN architecture.

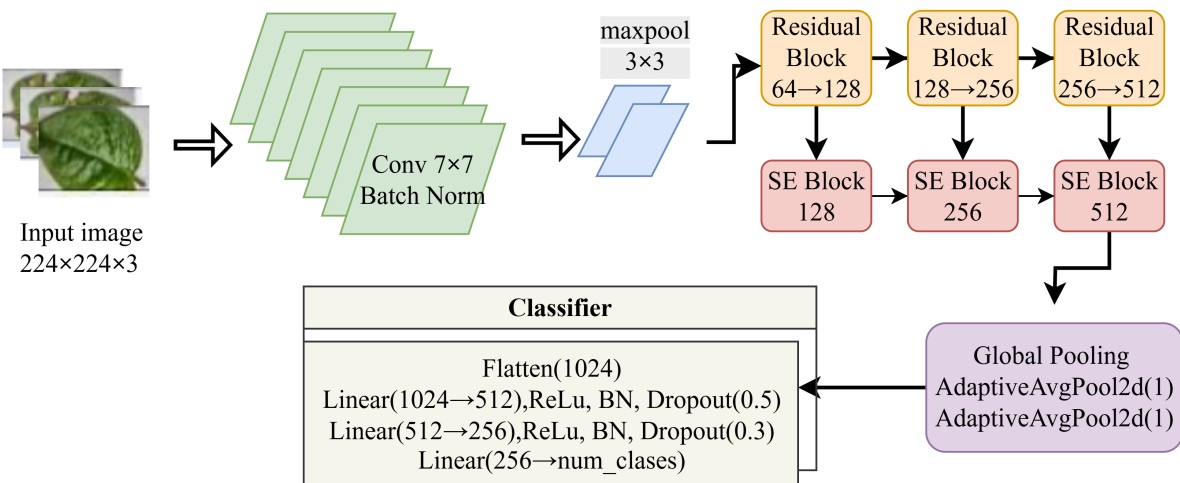

**Fig 5**. **Proposed Res-SENet architecture.**

Three Residual Blocks form the backbone of the proposed Res-SENet architecture, each consisting of a primary path and a skip connection. Residual connections allow the model to learn residual mappings; thus, deeper architectures are possible without performance degradation. Each residual block is supplemented with an SE module, which adaptively recalibrates channel-wise feature responses by exploiting the interdependencies among channels to enable the model to perform more accurate prioritization between the different channels.

Residual Block 1 contains two 3×3 convolutional layers with 128 output channels in the main path, separated by batch normalization and RELU activations. A skip connection is applied with a 1×1 convolution when the input dimensions do not match the output dimensions of the main path. The feature representation is refined with an SE block of 128 channels. The same goes for Residual Block 2, which has the same layout but 256 channels for the output on both the main path and the SE block. Lastly, Residual Block 3 has 512 output channels but follows the same architectural design.

The model utilizes a Global Pooling layer to pool spatial information. This layer is a concatenation of adaptive average pooling and adaptive max pooling outputs. The model captures complementary spatial features by combining these two pooling strategies, leading to a stronger feature representation.

The Res-SENet classifier then utilizes the pooled features through a few fully connected layers. Finally, the features are flattened and fed into a linear layer that transforms them into 512 dimensions, after which a ReLU activation is applied, followed by batch normalization and another dropout layer with a probability of 0.5. This step is followed by another linear layer that reduces to 256, then ReLU, batch normalization, and a dropout layer with $p$=0.3. The last linear layer accepts the features we got and maps them to the number of output classes, giving us classification logits. A softmax is then applied to the logits to get the class probabilities.

Our Spinach-ResSENet architecture was trained and run under the same, constant conditions. The AdamW optimizer was used, having the base learning rate set to $1 \times 10^{-4}$ and the weight decay of $1 \times 10^{-4}$, and it was trained for 30 epochs using a batch size of 32. Furthermore, the Cosine Annealing with Warm Restarts scheduler was applied. In the latter, $T_0 = 5$, $T_{mult} = 2$, and $\eta_{min} = 1 \times 10^{-6}$. In order to avoid overfitting, the early stopping technique was applied, with a patience of 5 epochs. The Cross-Entropy Loss label smoothing value was 0.1. In addition, the use of the Mixed Precision Training (AMP) technique for the improvement of the stability and the efficiency of the process was enabled. For the purpose of ensuring the reproduction of the experiments, the random state for the Python, NumPy, and Torch environments was locked to 42.

### 3.3.3 SpinachViT model.

The SpinachViT model, illustrated in Fig 6, is based on a ViT-Base style architecture adapted for leaf disease classification. Input images of size 224×224×3 are split into non-overlapping 16×16 patches, which are linearly projected into 768-dimensional patch embeddings. A learnable classification token (CLS) is prepended to the patch sequence, and learnable positional embeddings are added to preserve spatial information. The resulting sequence is then passed through a stack of 12 transformer encoder layers, each consisting of multi-head self-attention with 12 attention heads and a feed-forward multilayer perceptron (MLP) with expansion factor 4. Residual connections and layer normalization are applied around both the attention and MLP blocks to stabilize training.

After the final encoder layer, the CLS token is extracted, passed through layer normalization, and fed into a linear classification head that outputs logits for the three classes: Alternaria, straw mite, and healthy leaf. This configuration results in approximately 85.5M trainable parameters, providing a high-capacity transformer baseline for comparison against more lightweight CNN-based models.

The SpinachViT model was trained for 30 epochs using the AdamW optimizer (learning rate $3 \times 10^{-4}$, weight decay 0.05, mini-batch size 32) with a cosine annealing learning-rate scheduler. Image data were resized to 224×224 pixels and normalized using dataset-specific statistics before patch embedding. Cross-entropy loss with dropout regularization (0.1) was used for optimization. For comprehensive evaluation, Test-Time Augmentation (TTA) was applied using eight augmented views per test image, including horizontal flips, rotations of 90°, 180°, and 270°, and color jitter; predictions were averaged across views. Metrics for evaluation included accuracy, precision, recall, F1 score, confusion matrices, and macro-averaged AUC-ROC using a one-vs-rest scheme.

## 3.4 Self-supervised SimSiam (Vanilla ResNet-50)

SimSiam (Simple Siamese Networks) [25] was employed for self-supervised representation learning. Unlike contrastive methods that require negative pairs, very large batch sizes, or momentum encoders, SimSiam learns from positive pairs

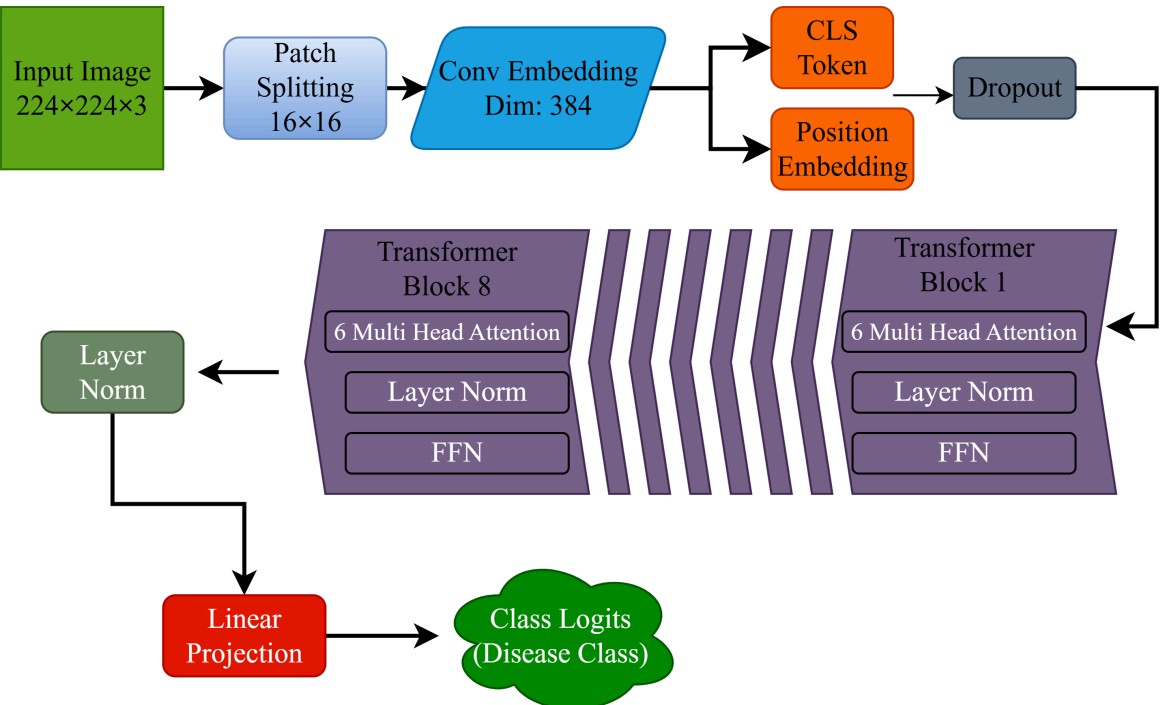

**Fig 6**. Proposed SpinachViT architecture.

only, constructed from two independently augmented views of the same image. A stop-gradient operation stabilizes training and prevents representational collapse, yielding a simple and effective self-supervised framework.

Two SimSiam pipelines were constructed in this study: (i) a Vanilla ResNet-50 encoder and (ii) a CBAM-enhanced ResNet-50. Each model underwent (1) self-supervised pretraining using unlabeled Malabar spinach leaf images, followed by (2) supervised fine-tuning using labeled images. Fig 7 illustrates the Vanilla ResNet-50 SimSiam framework.

**Stage I: SimSiam pretraining.** **Data processing and augmentation.** Unlabeled spinach leaf images were resized/cropped to $224 \times 224$ pixels and normalized using ImageNet statistics. For each input image, two independent augmented views were generated using a strong augmentation pipeline:

1. RandomResizedCrop (size 224, scale 0.2–1.0)
2. RandomHorizontalFlip ($p = 0.5$)
3. RandomApply(ColorJitter(0.4, 0.4, 0.4, 0.1), $p = 0.8$)
4. RandomGrayscale ($p = 0.2$)
5. GaussianBlur (kernel size 23, $\sigma \in [0.1, 2.0]$)
6. Tensor conversion and ImageNet normalization

This combination of spatial and photometric transformations exposes the encoder to diverse appearance variations and encourages view-invariant feature learning.

**Network architecture.** The encoder backbone was a ResNet-50 without its classification head (global average pooling output, 2048-D). Following the standard SimSiam design, a three-layer MLP projector was appended (`Linear–BN–ReLU → Linear–BN–ReLU → Linear–BN` with `affine=False`), along with a two-layer predictor MLP (`Linear–BN–ReLU → Linear`).

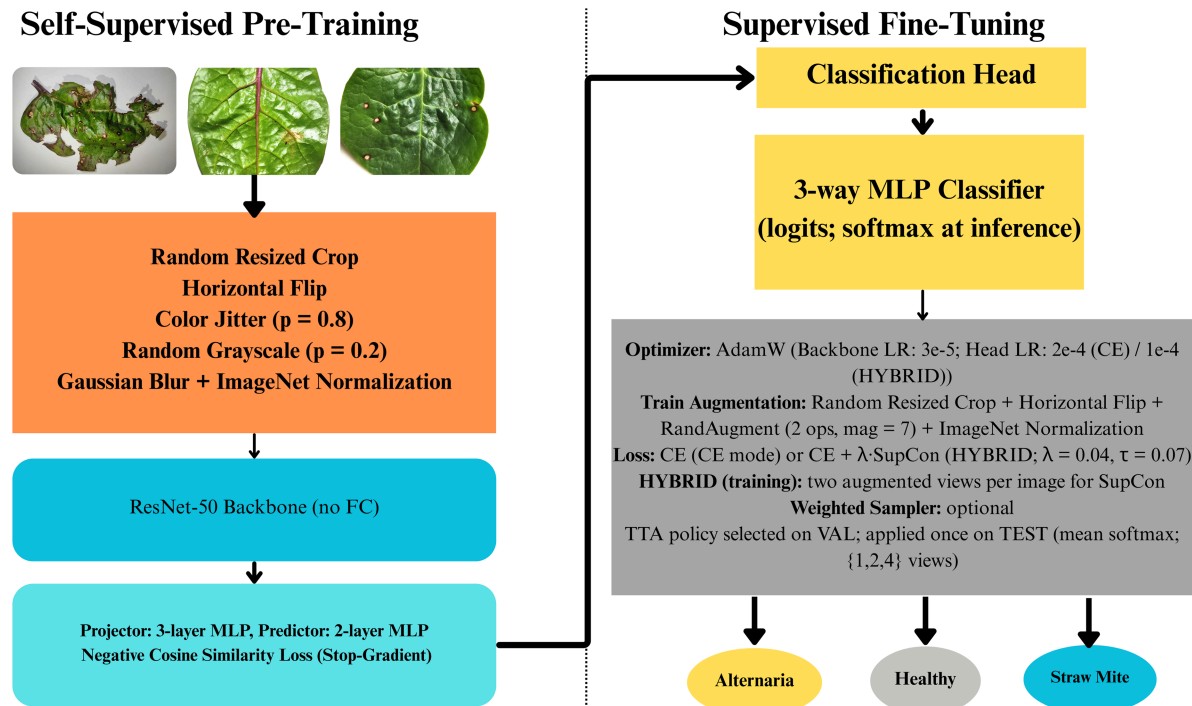

**Fig 7**. Proposed Vanilla ResNet-50 SimSiam pretraining and supervised fine-tuning pipeline.

**Objective function.** SimSiam minimizes a symmetric negative cosine similarity objective:

$$\mathcal{L} = \frac{1}{2}\left[\mathcal{D}(p_1, \text{stopgrad}(z_2)) + \mathcal{D}(p_2, \text{stopgrad}(z_1))\right], \tag{1}$$

where

$$\mathcal{D}(p, z) = -\frac{p}{\|p\|_2} \cdot \frac{z}{\|z\|_2}. \tag{2}$$

Here, $z$ is the projector output, $p$ is the predictor output, and `stopgrad` blocks gradients through the target representation, preventing collapse.

**Optimization.** Pretraining used SGD with momentum 0.9, weight decay $10^{-4}$, batch size 64, and a base learning rate linearly scaled as $\text{LR} = 0.05 \times \frac{\text{batch}}{256}$. A cosine learning-rate schedule was applied for 200 epochs. Mixed precision training (`torch.amp`) was used for efficiency.

**Stage II: Supervised fine-tuning. Network adaptation.** The SimSiam-pretrained ResNet-50 backbone weights were transferred to the supervised stage. The SimSiam projector/predictor were discarded, and a lightweight classification head was attached:

- a 512-unit fully connected layer,
- BatchNorm, ReLU, and 30% dropout,
- a final three-class linear output layer (logits).

**Supervised data augmentation and evaluation protocol.** Training used moderate augmentations: RandomResized-Crop (scale 0.7–1.0, ratio 0.9–1.1), RandomHorizontalFlip ($p = 0.5$), and RandAugment (2 operations, magnitude 7), followed by ImageNet normalization. Validation and test used a deterministic protocol: resize to 256 and center-crop to 224.

**Class imbalance handling.** Class imbalance was handled using an *optional* WeightedRandomSampler during mini-batch construction. When the sampler was enabled, Cross-Entropy was kept unweighted to avoid double-compensation; otherwise, class-weighted Cross-Entropy was used.

**Optimization and training strategy.** Fine-tuning ran for up to 60 epochs with early stopping (patience = 10) based on validation accuracy. AdamW was used with weight decay $10^{-4}$ and layer-wise learning rates, with a smaller learning rate for the backbone and a larger one for the classifier head. A cosine learning-rate scheduler was applied. Macro-F1 was recorded as a diagnostic metric (while early stopping was based on validation accuracy). Mixed precision training was used in both CE and hybrid settings.

**Hybrid loss configuration (optional).** Selected experiments used a CE-dominant hybrid objective combining Cross-Entropy and Supervised Contrastive Loss (SupCon). In the hybrid setting, **two augmented views per training image** were generated for SupCon, while Cross-Entropy was computed on one view. A 128-D projection head was used for contrastive embeddings in the supervised stage. The total loss was:

$$\mathcal{L}_{\text{total}} = \mathcal{L}_{\text{CE}} + \lambda \mathcal{L}_{\text{SupCon}}, \tag{3}$$

with $\lambda = 0.04$ and temperature $\tau = 0.07$. Unless otherwise stated, light mixup was enabled only in the hybrid setting ($\alpha = 0.05$) for the CE branch.

**Test-Time Augmentation (TTA).** For evaluation, predictions were averaged across multiple conservative test-time views using mean aggregation of softmax probabilities. The TTA policy was selected on the validation set from a small candidate set (1, 2, or 4 views) and then applied once to the test set. The 4-view policy consisted of:

- center-crop after resize 256,
- horizontal flip after resize 256,
- center-crop after resize 288,
- horizontal flip after resize 288.

Because the test set is relatively small and the TTA transforms are intentionally mild, TTA may yield performance comparable to a single forward pass; therefore, both single-crop and TTA results are reported.

### Algorithm 1 Pretraining and fine-tuning of the proposed Vanilla SimSiam-ResNet-50 model.

**Require:** Leaf image dataset: train, validation, test, and pretrain splits
**Ensure:** Best-performing model, evaluation metrics, and ablation summary
1: **Initialization:** Load dataset splits; define ResNet-50 backbone and heads; configure augmentations, losses, and optimizers.
2: **SimSiam Pretraining (Self-Supervised):**
3: **for** each epoch **do**
4: **for** each minibatch in pretrain set **do**
5: Generate two independently augmented views per image;
6: Forward: backbone → projector → predictor;
7: Compute SimSiam loss using symmetric negative cosine similarity with stop-gradient;
8: Backpropagate and update parameters using SGD;
9: **end for**
10: **end for**
11: Save pretrained backbone weights.
12: **Fine-Tuning (Supervised):**
13: Discard SimSiam heads; attach classification head (and SupCon projection if hybrid);
14: Load pretrained backbone weights into supervised model;
15: Optionally enable WeightedRandomSampler for class imbalance;
16: **for** each epoch **do**
17: **for** each minibatch in training set **do**
18: Apply supervised augmentations (RandAugment);
19: **if** Hybrid configuration **then**
20: Generate two augmented views per image for SupCon; compute CE on one view;
21: **end if**
22: Compute CE (or CE + $\lambda$SupCon) loss;
23: Backpropagate and update parameters using AdamW with layer-wise learning rates;
24: **end for**
25: Evaluate on validation set (early stopping monitors validation accuracy);
26: **if** validation accuracy does not improve for patience epochs **then**
27: Trigger early stopping;
28: **end if**
29: **end for**
30: Save best model weights.
31: **Evaluation:**
32: Load best model;
33: Compute single-crop test predictions;
34: Select a TTA policy on validation (1/2/4 views) and apply once on test;
35: Average softmax probabilities across TTA views;
36: Compute metrics: accuracy, macro ROC-AUC, confusion matrix, and diagnostic scores (e.g., macro-F1).
37: **Output:** Single-crop accuracy, TTA accuracy, macro ROC-AUC, and ablation summary.

### 3.5 Channel-spatial attention SimSiam (CBAM-ResNet-50)

**Stage I: SimSiam pretraining** Pipeline: The SimSiam pretraining pipeline for the CBAM-ResNet-50 model followed the same procedure as the one for the vanilla ResNet-50 encoder. Unlabeled and non-augmented images were used, from which two strongly augmented views were generated using random resized cropping, horizontal flipping, color jittering, grayscale conversion, Gaussian blur, and ImageNet normalization. These dual views provided the positive pairs necessary for the SimSiam objective.

Network Architecture: As shown in Fig 8, the CBAM-ResNet-50 architecture extends the standard ResNet-50 by inserting a Convolutional Block Attention Module (CBAM) into every residual bottleneck block. Each CBAM block consists of:

**3.5.1 Channel attention.** Achieves global information by average pooling and max pooling, and then it uses a shared MLP implemented with 1×1 convolutions (followed by sigmoid activation).

**3.5.2 Spatial attention.** Aggregates spatial cues using average and max pooling along the channel dimension, followed by a 7×7 convolution and sigmoid activation.

These attention mechanisms recalibrate the feature maps so that the encoder emphasizes disease-relevant leaf regions while suppressing irrelevant background texture.

CBAM modules operate throughout the ResNet-50 bottleneck hierarchy (all residual bottleneck blocks across the backbone stages). A two-layer MLP projector and a two-layer SimSiam predictor were attached exactly as in the vanilla SimSiam pipeline. During pre-training, BatchNorm layers inside the backbone were kept in evaluation mode and frozen, ensuring stable representations during self-supervised training.

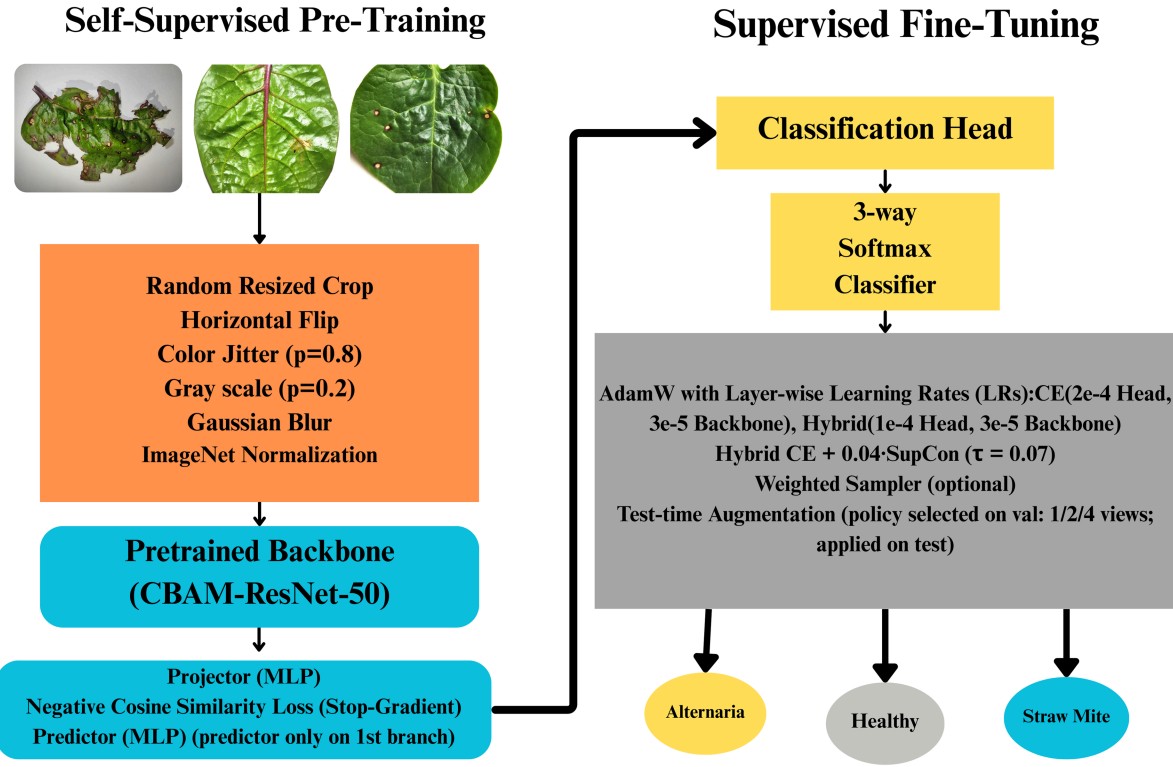

**Fig 8**. **Proposed CBAM-ResNet-50 framework.**

Optimization was performed using SGD with momentum 0.9, weight decay $10^{-4}$, and a cosine learning rate schedule with SimSiam-style learning-rate scaling (batch size 64). Based on negative cosine similarity between prediction and stop-gradient target projections, the loss function was identical to that used for the vanilla SimSiam model.

**Stage II: Supervised fine-tuning.** Network Adaptation: The pretrained CBAM-ResNet-50 backbone was loaded from the SimSiam checkpoint, and a classification head was used as:

- 512-unit fully connected layer
- BatchNorm, ReLU, Dropout (0.30)
- Final 3-class linear classifier

Fine-tuning incorporated AdamW optimization with layer-wise learning rates into two groups (backbone and head). For CE, the backbone learning rate was $3 \times 10^{-5}$ and the head learning rate was $2 \times 10^{-4}$. For Hybrid, the backbone learning rate remained $3 \times 10^{-5}$ and the head learning rate was $1 \times 10^{-4}$. A cosine annealing scheduler and early stopping were applied.

Mixup-based training was applied in the Hybrid setting (default $\alpha = 0.05$ in code), and RandAugment was used in the training pipeline to improve robustness and regularization.

Both cross-entropy and hybrid loss (CE + supervised contrastive) configurations were evaluated. In the Hybrid configuration, the objective used in code is additive:

$$\mathcal{L}\text{total} = \mathcal{L}\text{ce} + \lambda\mathcal{L}_{\text{sup}}, \quad \lambda = 0.04, \tag{4}$$

with temperature $\tau = 0.07$.

This loss encourages features from the same disease class to cluster more tightly in embedding space, complementing the attention mechanisms in CBAM.

### 3.6 Evaluation protocol for both SimSiam frameworks

Evaluation was conducted using Test-Time Augmentation (TTA) after fine-tuning. The TTA policy was selected on the validation set from 1, 2, 4 views and then applied once on the test set. The views include center-crop evaluation at 256 resize, a horizontal-flip counterpart, and an additional resize level (288) in the 4-view setting. Final predictions were computed by averaging softmax probabilities across the selected augmented views. The reported metrics include:

- Overall accuracy
- Class-wise F1-scores
- Macro ROC-AUC
- Confusion matrix

Expected Calibration Error (ECE) was additionally reported in the robustness ablation experiments (noise training) with post-hoc temperature scaling to ensure calibrated probabilities.

### 3.7 Swin transformer

SwinV2, a hierarchical vision transformer architecture [26], was employed for Malabar spinach leaf disease classification. Through shifted windowed self-attention, SwinV2 constructs multi-scale feature hierarchies while maintaining linear computational complexity with respect to image size. Fig 9 illustrates the configuration incorporated in this study.

## Algorithm 2 Pretraining and fine-tuning of the proposed CBAM-ResNet-50 with SimSiam.

**Require:** Leaf image dataset: train, validation, test, and pretrain splits
**Ensure:** Best-performing model, evaluation metrics, and ablation summary
 **Initialization:**
2: Load all dataset splits (train, val, test, pretrain);
 Define CBAM-ResNet-50 backbone;
4: Configure data augmentations (strong SimSiam augs for pretrain; RandAugment for fine-tune), loss functions (CE / CE+SupCon), and optimizer.
 **SimSiam Pretraining (Self-Supervised):**
6: **for** each epoch **do**
 **for** each minibatch in pretrain set **do**
8: Generate two augmented views per image;
 Pass both views through CBAM-ResNet-50 backbone and projection head;
10: Pass outputs through prediction head;
 Compute SimSiam loss using negative cosine similarity;
12: Backpropagate and update parameters;
 **end for**
14: **end for**
 Save pretrained weights (backbone + SimSiam heads).
16: **Fine-Tuning (Supervised):**
 Load pretrained backbone weights;
18: Attach classification head;
 **for** each epoch **do**
20: **for** each minibatch in training set **do**
 Apply augmentations (RandAugment), and mixup in the Hybrid setting;
22: Perform forward pass and compute classification loss (CE or CE+SupCon);
 Backpropagate and update parameters using AdamW;
24: **end for**
 Evaluate on validation set;
26: **if** validation accuracy does not improve for patience epochs **then**
 Trigger early stopping;
28: **end if**
 **end for**
30: Save best model weights.
 **Evaluation and Ablation:**
32: Load best model;
 Select TTA policy on validation from 1,2,4 views and apply to test;
34: Compute metrics: accuracy, macro ROC-AUC, and confusion matrix;
 Perform noise ablation (Gaussian / salt-pepper / both) and report robustness; apply temperature scaling for calibrated ECE reporting.
36: **Output:** Final accuracy, TTA accuracy, ROC-AUC, and ablation statistics.

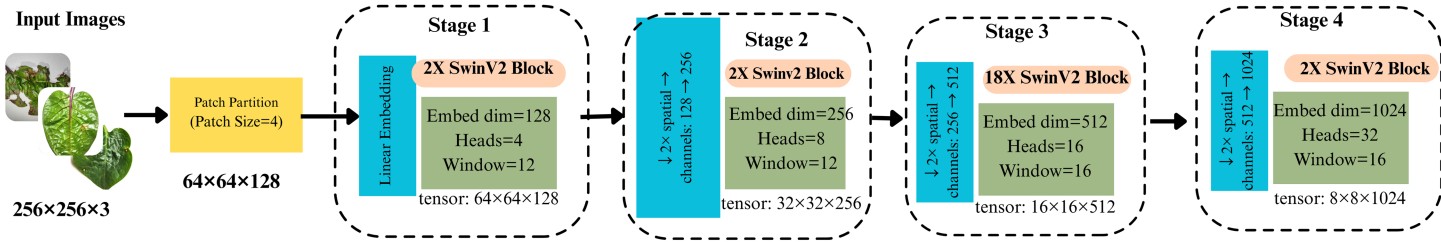

**Fig 9**. Applied swin transformer architecture.

**Patch Embedding and Hierarchical Stages:** Input leaf images of 256×256×3 were initially partitioned into non-overlapping 4×4 patches, generating a 64×64 grid of patch tokens. These tokens were processed through four hierarchical stages with patch merging between stages. Shifted windows were applied between successive layers, enabling cross-window information flow and facilitating global contextual reasoning over leaf surfaces. In this work, the SwinV2-Base hierarchy followed a depth configuration of [2, 2, 18, 2] with stage-wise embedding dimensions (128, 256, 512, 1024) and attention heads (4, 8, 16, 32); the window size was 12 in Stages 1–2 and 16 in Stages 3–4 (Fig 9). Patch merging halves the spatial resolution at each transition (64→32→16→8) while doubling the channel dimension.

**Classification Head:** Global average pooling was used to combine the final spatial feature map into a feature vector, which was then passed through a fully connected classifier to produce logits for the three disease categories: Alternaria, straw mite, and healthy leaf.

**Pre-training and Optimization Objective:** The SwinV2 model was implemented using the `timm` library. Two initialization settings were considered using the same SwinV2-Base architecture: (i) *random initialization (scratch)* and (ii) *pretrained initialization* using `swinv2_base_window12to16_192to256.ms_in22k_ft_in1k`. Fine-tuning used label-smoothed cross-entropy loss with smoothing factor 0.1.

**Training Configuration:** The controlled SwinV2-Base configuration `swinv2_base_window12to16_192to256` was used with:

- Optimizer: AdamW
- Learning rate: $1 \times 10^{-4}$
- Weight decay: 0.05
- Epochs: up to 30 (with early stopping)
- Batch size: 32
- Mixed precision training: enabled
- Early stopping: patience 7

**Evaluation Protocol (Strict):** To avoid center-bias from cropping, validation and testing used a strict resize protocol: all images were resized directly to 256×256 without center cropping.

**Performance Overview:** Under the strict evaluation protocol, the SwinV2-Base model achieved strong performance. The pretrained initialization (`ms_in22k_ft_in1k`) improved results compared to scratch, demonstrating the benefit of large-scale pretraining when fine-tuned on the Malabar spinach dataset.

### 3.8 Explainable AI (XAI)

In the study, the following Explainable AI (XAI) techniques were employed to enhance model interpretability.

- **Gradient-weighted Class Activation Mapping (Grad-CAM)** [27] was used to provide visual explanations for the predictions made by the applied deep learning models by highlighting class-discriminative regions in input images. The process comprised three steps: (a) Forward Pass: passing an input image through the trained network and obtaining feature maps from the target convolutional layer (usually the last conv layer before classification); (b) Gradient Computation: calculating gradients of the predicted class score concerning the feature maps by using backpropagation; and (c) Heatmap Generation: computing channel-wise importance weights by global average pooling the gradients, then linearly combining the feature maps with these weights to yield a coarse localization map allowing intuitive explanations.
- **Gradient-weighted Class Activation Mapping++ (Grad-CAM++)** is a newer visual explanation method that is an improvement over the regular Grad-CAM approach by including higher-order gradient information. For situations where an object or a feature needs to be highlighted more than once, this improvement allows for locating important areas in an image more accurately.

 

Grad-CAM++ is an improvement over Grad-CAM by employing higher-order gradients for more precise localization of salient image regions. It weights activations based on both first and second-order derivatives, rather than mere gradient averaging, to allow more accurate highlighting of multiple discriminative regions. This technique generates sharper visual explanations without modifying the original model architecture, and it works well, especially for complicated scenes with several relevant features. The technique accomplishes this by an improved weighting scheme that captures the spatial importance of features in a better way while still being computationally light and easy to implement.

- **Layer-wise Class Activation Mapping (LayerCAM)** [28] was utilized to exhibit fine-grained visual explanations for the predictions of the CNN by highlighting class-relevant areas in the input images through the utilization of spatial attention obtained from activation maps. It involved a three-step process: (a) Forward Pass – forward passing an input image through the trained network and retrieving activation maps from the target convolutional layer; (b) Gradient Computation – calculating gradients of the predicted class score against the activation maps through backpropagation; and (c) Heatmap Generation – applying ReLU to the gradients and element-wise multiplication with the corresponding activation maps to retrieve spatially accurate attention maps, which were aggregated to create a fine-resolution localization map that better indicates class-discriminative areas than Grad-CAM.

## 4 Results and performances

This section presents the findings and results of the various deep learning methodologies applied in this work for Malabar spinach leaf disease classification, starting with our analysis of the performance of pretrained models, including Efficient-Net, ResNet, etc. Proceeding from the custom model architectures, i.e., SpinachCNN, the Spinach Res-SENet model, and SimSiam models, this section also presents a close analysis and comparative evaluation of all models employed in our work.

Based on the accuracy, precision, recall, and F1 score, Table 1 illustrates a detailed evaluation of the pretrained models for an adequate comparison. The EfficientNetB0 model accomplishes the highest accuracy of 98%. ResNet-50 demonstrates slightly lower accuracy (97.7%) but powerful performance for all metrics. MobileNetV2 attains a good balance, with 96.7% accuracy and high precision/recall.

### 4.1 Performance of Spinach-CNN

The performance of the custom SpinachCNN model in terms of training and validation on the Spinach dataset is demonstrated in Fig 10, where the trends of loss and accuracy are presented over 30 epochs. The loss curve shows a steady decrease in training and validation losses, with the training loss starting from above 1.1 and gradually decreasing to approximately 0.5. Despite some oscillations depicting slight variations in the model's generalization capacity, the validation loss follows the same trend. The accuracy plot illustrates that the model learns effectively, as the training accuracy improves from approximately 50% to nearly 90% over the epochs. Reaching 80% validation accuracy within five epochs, the model demonstrated excellent performance and maintained a steady level of around 90%. The model is efficient, has high accuracy and good feature extraction from the Spinach dataset, and demonstrates strong generalization capability.

**Table 1**. **Performance comparison of four deep learning models.**

| Models | Accuracy (%) | Precision (%) | Recall (%) | F1 score (%) |
|---|---|---|---|---|
| EfficientNetB0 | 98.0 | 98.0 | 98.0 | 98.0 |
| ResNet-50 | 97.7 | 97.6 | 97.6 | 97.6 |
| MobileNetV2 | 96.7 | 96.6 | 96.7 | 96.6 |
| ViT-B/16 | 91.0 | 91.0 | 91.0 | 91.0 |

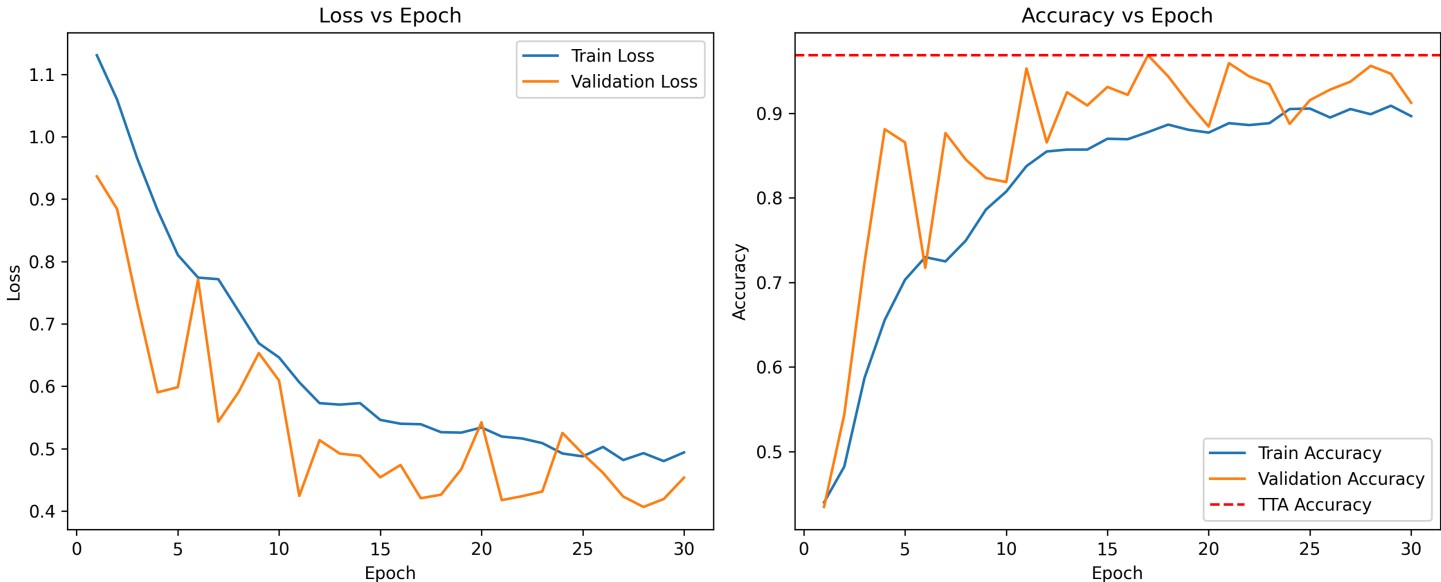

**Fig 10**. Accuracy and Loss vs. epoch for SpinachCNN.

## 4.2 Performance of Spinach-ViT

The performance of the Vision Transformer-based SpinachViT model on the employed spinach dataset is presented in Fig 11, which includes training and test loss and accuracy trends across 30 epochs. As observed from the loss plot, a smooth drop in the training loss from above 0.8 to near zero indicates good model optimization. As the test loss varies across wide ranges during training, the test loss exhibits some instability in terms of generalization. The test loss starts to

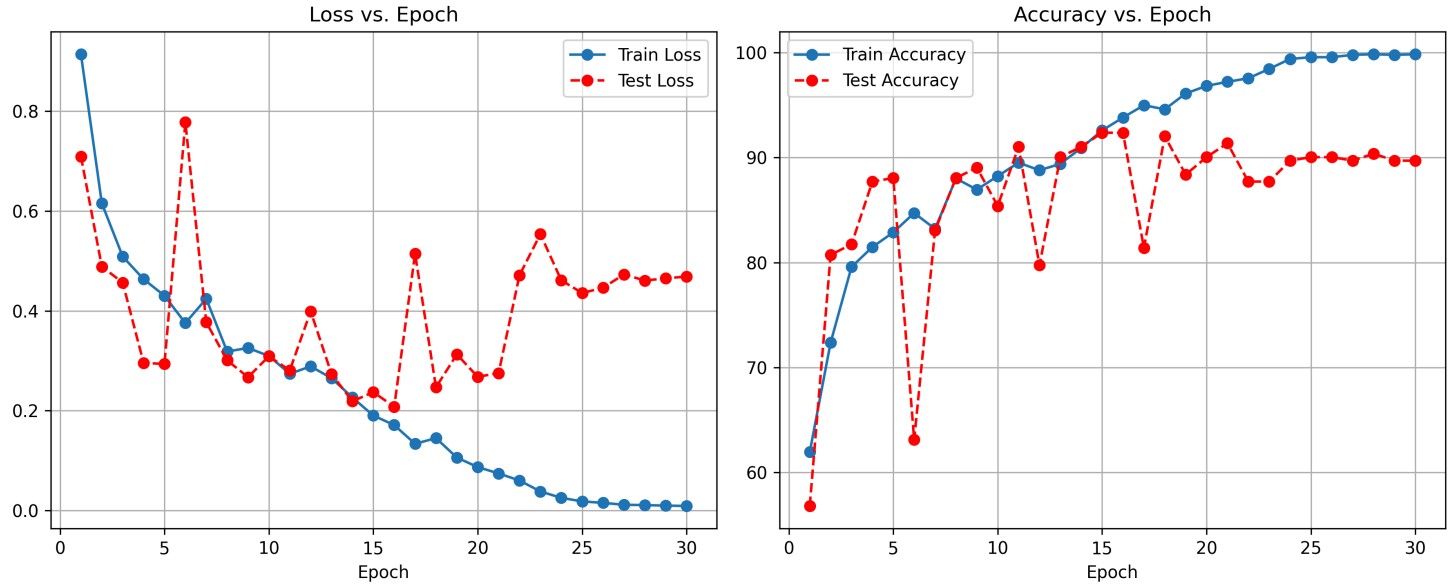

**Fig 11**. Accuracy and Loss vs. epoch for Spinach-Vit model.

increase approximately after 20 epochs, even though it initially shows a decreasing trend, potentially indicating the onset of overfitting.

With training accuracy reaching close to 100% towards the final epochs, the plot for accuracy replicates significant performance. In the early stages, the test accuracy exhibits quick improvement to over 80% within the first few epochs. However, it also demonstrates significant variability, highlighting some irregularity in the ability of the model to generalize. The ViT model shows great feature learning abilities on the Spinach dataset and is highly accurate, but needs more fine-tuning to achieve optimal stability.

### 4.3 Performance of Spinach-Res-SENet

The performance of the Spinach Res-SENet model over 30 epochs is shown in Fig 12. Both the training loss and the validation loss decline steadily in the early epochs, as observed from the loss plot. After around 10 epochs, validation loss converges with some fluctuations, suggesting a possible convergence, but training loss continues to decline, suggesting continuous model fitting improvement.

Both validation and training accuracy increase during the initial epochs. The model achieves a validation accuracy of close to 90% at the completion of training, indicating that it performs effectively. Although there are some minimal changes in the later phases, no indications of severe overfitting or underfitting are present, implying that the model capacity and regularization are well-balanced.

Fig 13 shows the classification performance of our Spinach Res-SENet model, with only negligible misclassifications. The classification for Class 1 is perfect. Class 0 has been mainly classified into Class 1 with slight misclassifications, while Class 2 has been perfectly predicted.

### 4.4 Performance of Spinach-SimSiam-ResNet-50

Fig 14 summarizes the test-set performance of the proposed Spinach-SimSiam-ResNet-50 with standard Cross-Entropy fine-tuning (CE). The confusion matrices show that most errors arise from confusion between *Alternaria* and *Healthy Leaf*, while *straw_mite* remains consistently well-separated.

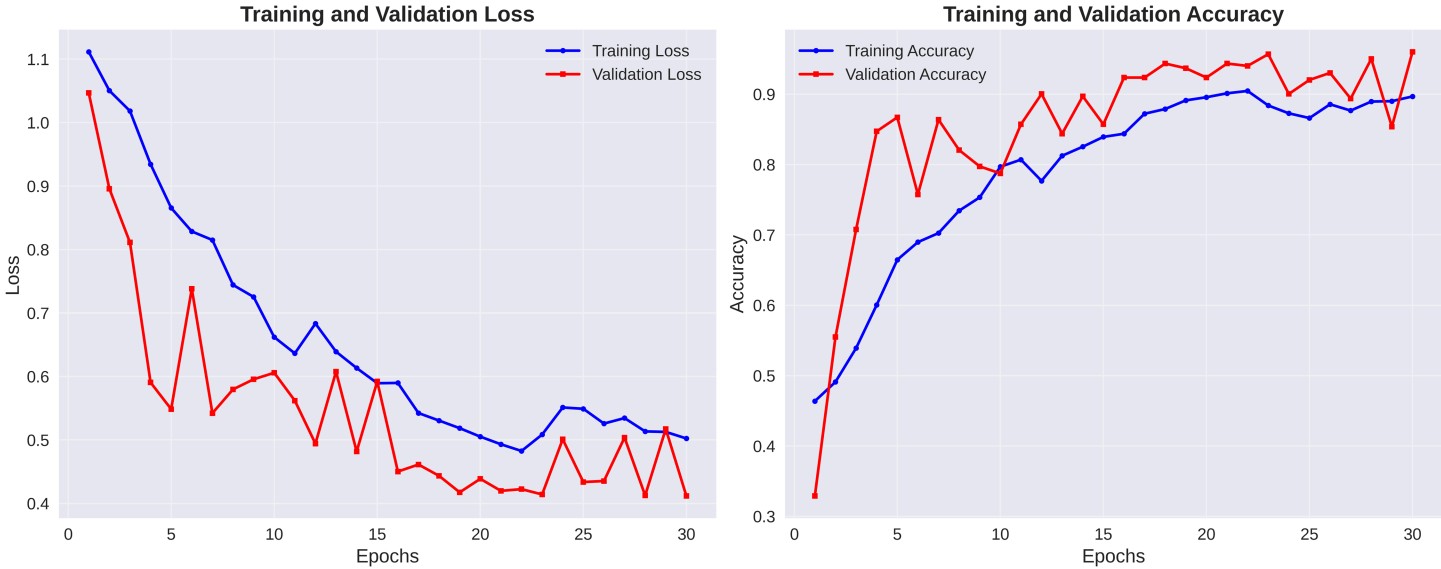

**Fig 12**. **Loss and Accuracy vs. epoch of the Spinach-Res-SENet model.**

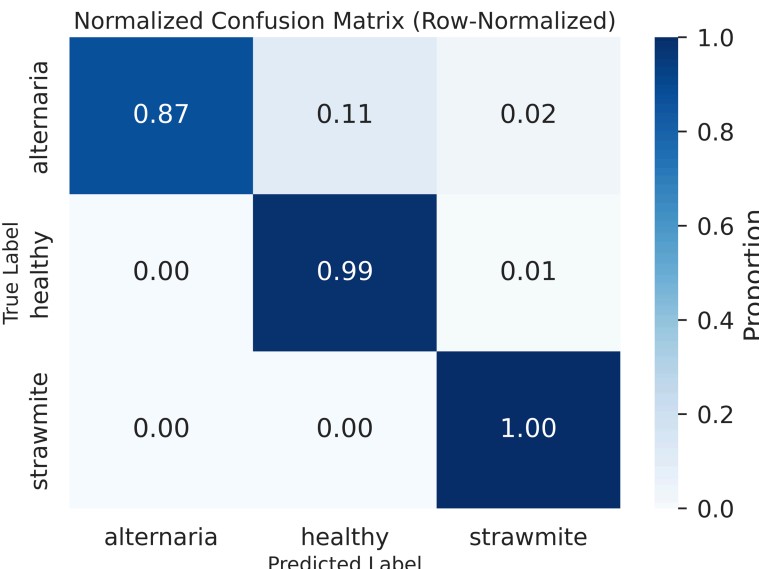

**Fig 13**. **Confusion Matrix of the Spinach-Res-SENet model.**

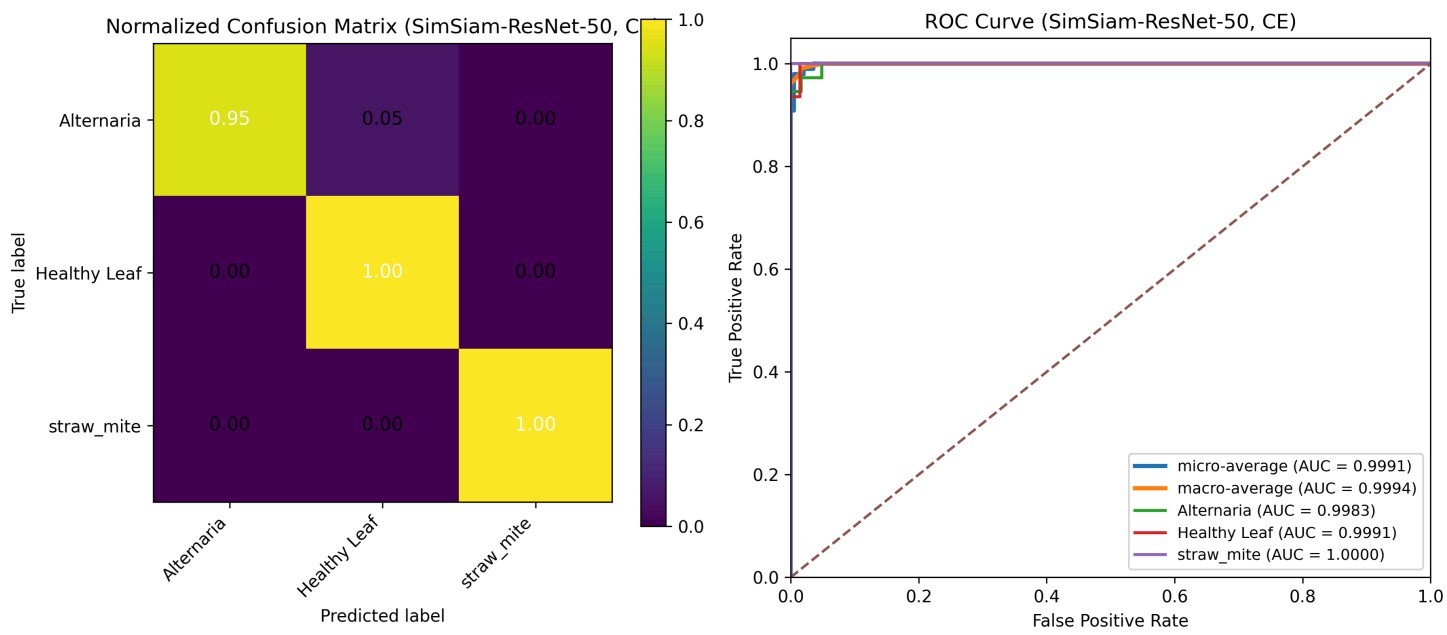

**Fig 14**. **Test-set performance of Spinach-SimSiam-ResNet-50 with CE fine-tuning.**

Under CE fine-tuning (Fig 14 the model almost correctly classified *Alternaria* samples (0.95), and achieved perfect classification for both *Healthy Leaf* (1.00) and *straw_mite* (1.00). The ROC analysis further confirms near-saturated separability, with macro- and micro-average AUC values of 0.9994 and 0.9991, respectively, and per-class AUC reaching 1.0000 for *straw_mite*.

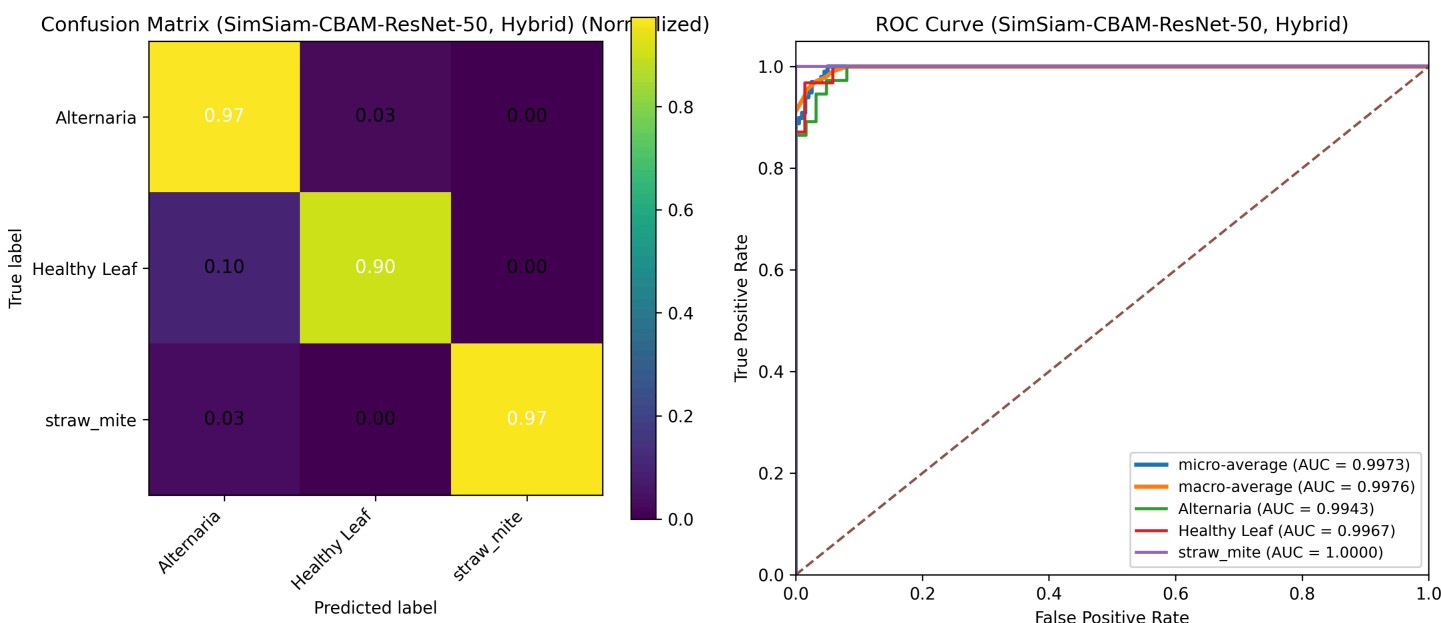

Under Hybrid fine-tuning, the model almost correctly classified *Alternaria* samples (0.92), *Healthy Leaf* samples (0.97), and again achieved perfect classification for *straw_mite* (1.00). While the Hybrid configuration increases the emphasis on feature-level class separation during training, the ROC curves show lower AUC values than CE in this particular checkpoint (macro AUC = 0.9895; micro AUC = 0.9855), with *straw_mite* remaining unchanged at AUC = 1.0000.

### 4.5 Performance of Spinach-SimSiam-CBAM-ResNet-50

Fig 15 shows the test-set performance of Spinach-SimSiam-CBAM-ResNet-50 under Hybrid fine-tuning (CE $+\lambda \cdot$ SupCon, with $\lambda = 0.04$ and $\tau = 0.07$). Nearly all samples are correctly classified, with the few errors mainly occurring between *Alternaria* and *Healthy Leaf*; *straw_mite* remains consistently well-separated. The ROC curves further confirm strong class separability, with macro- and micro-average AUC values close to 1.0 for both objectives.

### 4.6 Performance of swin transformer

Fig 16 evaluates the performance of the Swin Transformer model on the test dataset. The confusion matrix shows strong classification performance across all three classes. Under the strict evaluation protocol (direct resize to 256×256 with no center crop), the pretrained SwinV2-Base model achieved high accuracy and macro F1-score, confirming its robust discriminative capability for Malabar spinach leaf disease classification.

### 4.7 Comparative analysis and ablation study

Using the fixed training, validation, and test splits described in Sect 3.1, all robustness ablation experiments are conducted under the same base pipeline as the main setting (identical backbone, optimizer schedule, and training budget). The only controlled variable is the injection of synthetic noise into the *training images only* through the preprocessing/augmentation pipeline. Evaluation is always performed on the original *clean* validation and test sets.

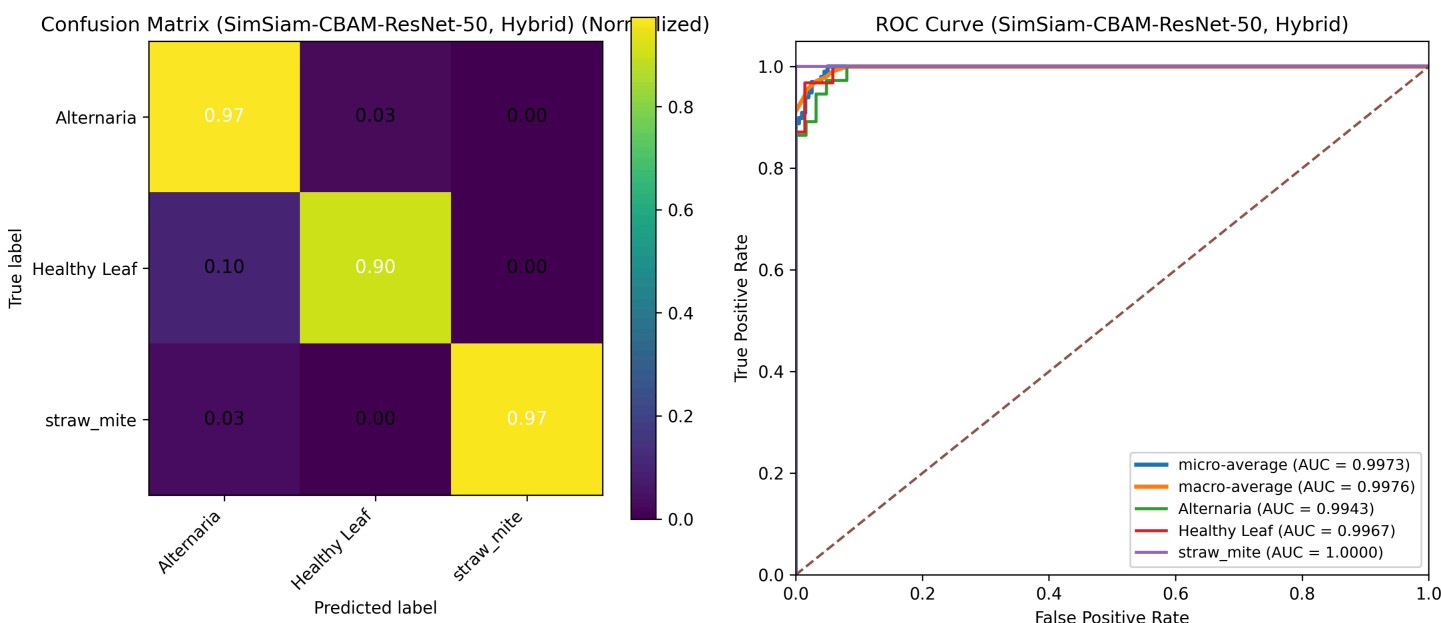

**Fig 15**. **Test-set performance of Spinach-SimSiam-CBAM-ResNet-50 under hybrid fine-tuning.**

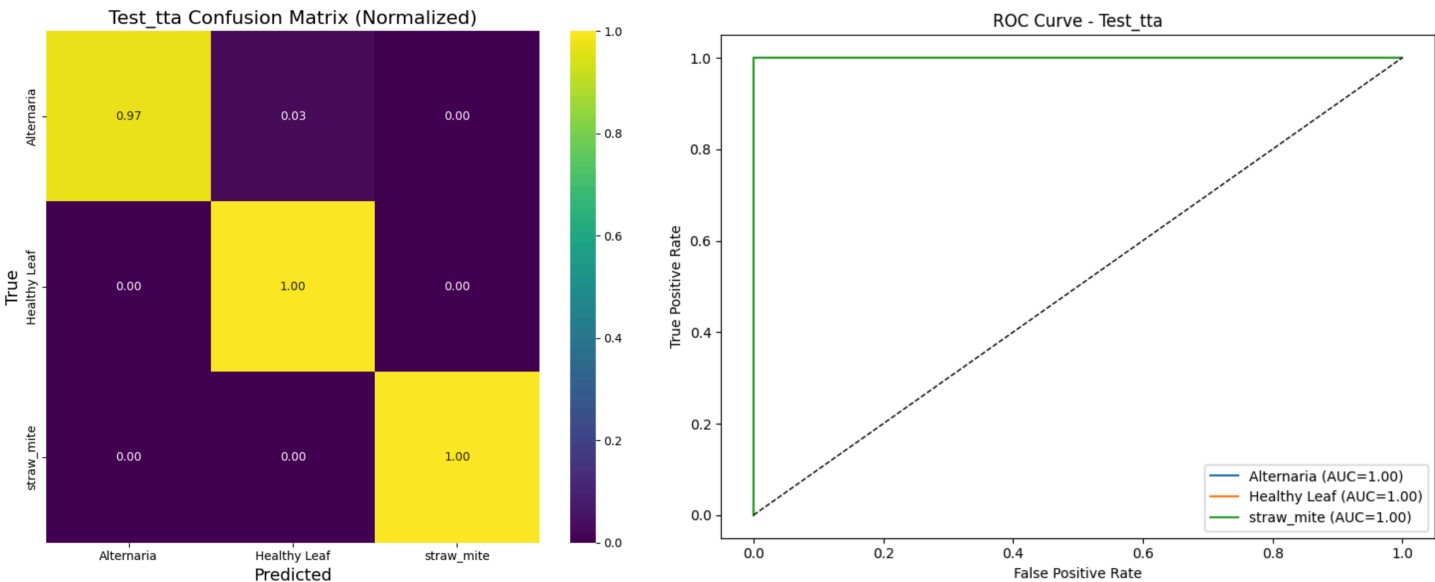

**Fig 16**. **Confusion matrix for the applied swin transformer.**

Four noise conditions are studied:

- **No Noise:** Standard augmentation without additional noise.
- **Gaussian Only:** Additive Gaussian noise is applied to every training image.
- **Salt-and-Pepper Only:** Salt-and-pepper noise is applied to every training image.
- **Both Noises:** Gaussian noise and salt-and-pepper noise are applied sequentially to each training image.

As illustrated in Fig 17, noise augmentation is applied exclusively during training to assess whether the model can learn robust representations under perturbations while maintaining generalization to clean inputs. All experiments are performed on the **SimSiam-CBAM-ResNet-50** architecture, initialized from the same SimSiam in-domain pretraining (Sect 3.5) and fine-tuned using the cross-entropy objective (Sect 3.4). For fairness and reproducibility, we reuse the best-performing CE seed identified in our main three-seed evaluation and keep the remaining hyperparameters unchanged.

**Evaluation metrics.** We report **Test Accuracy**, **Macro ROC-AUC**, and **Expected Calibration Error (ECE)**. Since calibration can be significantly improved post hoc without affecting classification accuracy, we report **temperature-scaled ECE** (ECE(TS)) computed by fitting a single temperature parameter on the validation logits and evaluating calibration on the test set.

Table 2 shows that SimSiam-CBAM-ResNet-50 maintains strong discriminative performance under synthetic perturbations introduced during training. While both single-noise settings produce only a modest reduction in accuracy relative to the clean-training baseline, the combined noise setting yields the largest degradation, indicating additive difficulty when multiple corruption types are applied concurrently. Macro ROC-AUC remains high across all conditions ($\geq 0.992$), suggesting that class-agnostic separability is preserved even when the training distribution is corrupted. In terms of reliability, temperature scaling substantially improves calibration, keeping ECE(TS) within a moderate range (3.31–6.36%), even for the most challenging combined-noise condition.

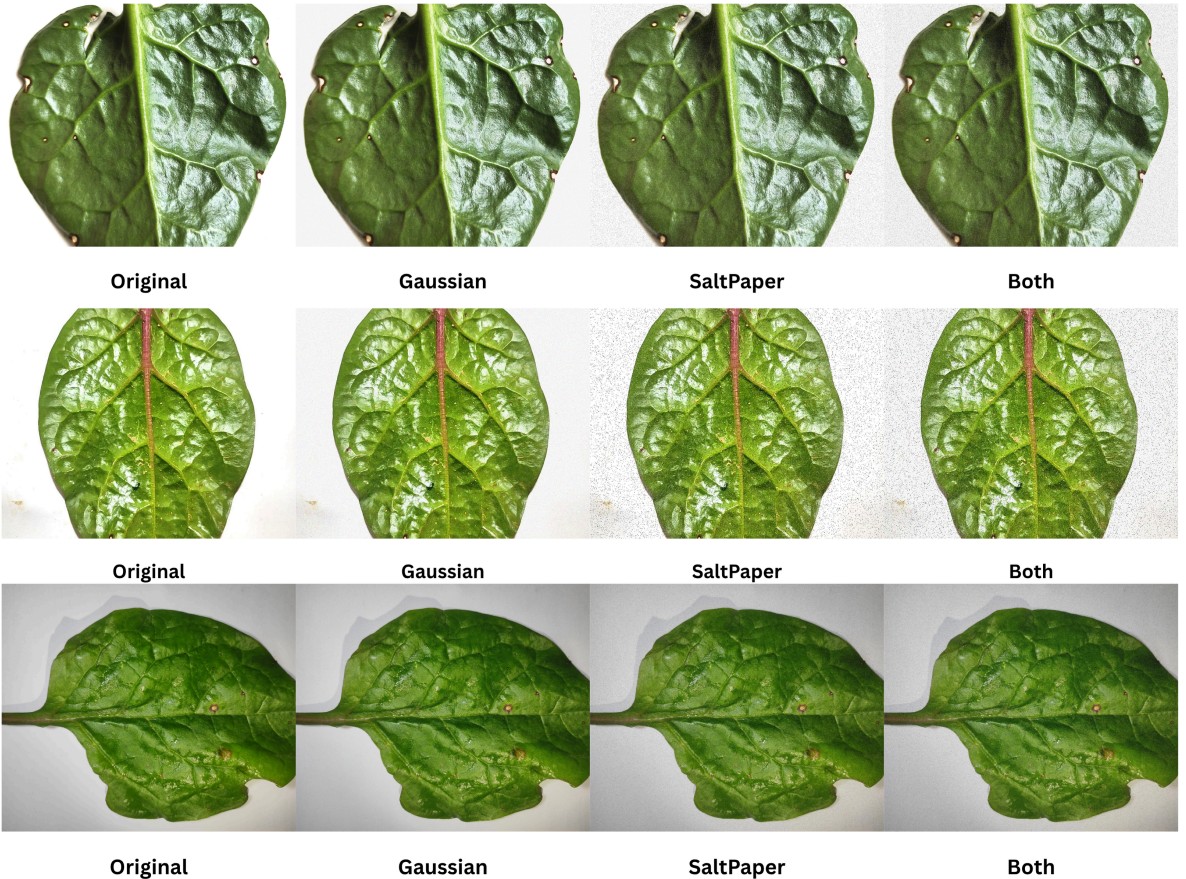

**Fig 17**. **Examples of noise-augmented training images used in the robustness ablation study.**

**Table 2**. **Ablation study: robustness of SimSiam-CBAM-ResNet-50 to synthetic noise (training-only noise).**

| Noise Condition | Test Accuracy (%) | Macro ROC-AUC | ECE(TS) (%) |
|---|---|---|---|
| No Noise | 97.98 | 0.9986 | 5.02 |
| Gaussian Only | 95.96 | 0.9948 | 3.31 |
| Salt-and-Pepper Only | 95.96 | 0.9977 | 5.97 |
| Both Noises | 93.94 | 0.9920 | 6.36 |

*Note:* Noise is applied during training only; validation and testing are performed on clean samples. ECE(TS) denotes Expected Calibration Error after post-hoc temperature scaling (lower is better).

## 4.8 Visualizations and qualitative results

XAI techniques were applied to the applied SimSiam-CBAM-ResNet-50 model, which demonstrated excellent classification accuracy on the test set, with an average confidence score of around 94.4% ($\pm$3.8%). Class-wise analysis revealed that predictions for straw mite exhibited the highest mean confidence, about 96.3%$\pm$2.5%, followed by healthy 95.3%$\pm$3.6%, and Alternaria 90.1%$\pm$2.4%. Analysis using Grad-CAM, Grad-CAM++, and Layer-CAM consistently highlighted biologically important regions in all samples, with Grad-CAM++ showing superior localization (mean intensity: 0.75) compared to Grad-CAM (0.72) and Layer-CAM (0.68). Fig 18 compares CAM visualizations for a straw mite-infected leaf (96.4% confidence). All methods highlight mite damage patterns, with Grad-CAM++ showing sharper lesion localization

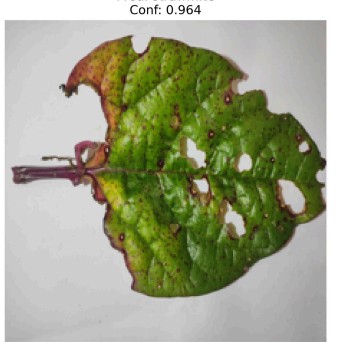
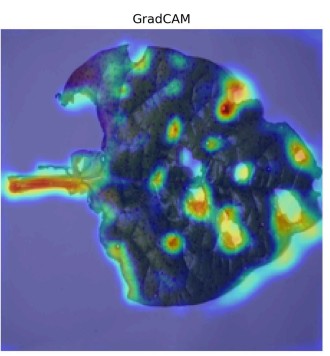
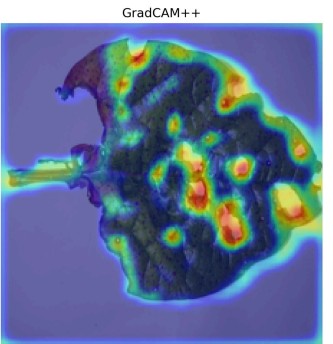
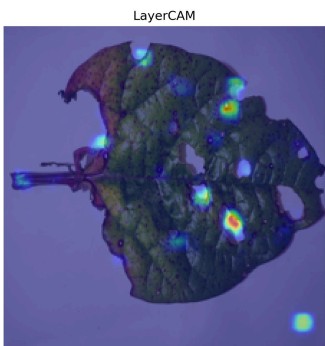

Original
Pred: strawmite
Conf: 0.964

GradCAM

GradCAM++

LayerCAM

**Fig 18**. **XAI Explainability Heatmap with Grad-CAM, Grad-CAM++, and LayerCAM.**

than Grad-CAM's broader activations, while Layer-CAM reveals finer details. The model's decision was validated by the consistent focus on biologically relevant features.

According to Table 3, the comparative performance shows several salient findings across model architectures. The Spinach-CNN baseline recorded test accuracy of 91.00% and TTA accuracy of 96.68% using 5.49M parameters, indicating that even a lightweight CNN can benefit from test-time augmentation but remains limited in single-crop generalization. Spinach-ResSENet performed best among the compact ResNet-based variants, achieving 96.01% test accuracy and 95.35% TTA accuracy with 5.53M parameters, showing that squeeze-and-excitation (SE) attention improves feature quality without increasing model size substantially. The Spinach-ViT model, trained from scratch, produced comparatively lower scores (90.7% test, 91.3% TTA) despite its larger capacity (85.5M parameters), which is consistent with transformer training being more data-hungry under limited in-domain supervision. Unless otherwise specified, confusion matrices and example ROC curves in this section are plotted for the best-performing random seed of each configuration, whereas Table 3 reports mean $\pm$ standard deviation over three seeds. This explains small numerical differences between some per-class results in the figures and the aggregate statistics in the table.

For the self-supervised pipelines, SimSiam-ResNet-50 shows improved results over the simple CNN baseline when averaged over seeds. Under CE fine-tuning, SimSiam-ResNet-50 achieved $91.92 \pm 5.34\%$ test accuracy with macro ROC-AUC $0.9921 \pm 0.0071$. The Hybrid objective (CE $+,\lambda\cdot$SupCon) increased mean accuracy to $94.61 \pm 1.54\%$, although its macro ROC-AUC in this protocol was $0.9850 \pm 0.0026$. After incorporating channel-spatial attention, SimSiam-CBAM-ResNet-50 (CE) delivered the strongest overall performance among the SimSiam-based models, reaching $97.31 \pm 1.17\%$ test accuracy with identical TTA accuracy ($97.31 \pm 1.17\%$) and a near-saturated macro ROC-AUC of $0.9983 \pm 0.0003$ at $27.08 \pm 0.00$M parameters. This highlights that CBAM attention provides a consistent gain when paired with in-domain SimSiam pretraining and standard CE fine-tuning.

The SimSiam-CBAM-ResNet-50 (Hybrid) configuration shows a modest reduction in accuracy compared to its CE counterpart, achieving $95.29 \pm 0.58\%$ test accuracy (TTA $95.29 \pm 0.58\%$) with macro ROC-AUC $0.9967 \pm 0.0006$. Although supervised contrastive learning (SupCon) often improves generalization, its effect depends strongly on dataset size, class structure, and batch composition. In our setting, the hybrid objective can reasonably yield lower accuracy for several reasons. First, the hybrid loss optimizes a representation geometry rather than the evaluation metric. The CE term drives the classifier toward decision-boundary separation, whereas SupCon encourages tighter intra-class clustering and larger inter-class angular margins. For fine-grained leaf disease imagery, where Alternaria and healthy leaves may share subtle texture cues, excessive compactness can unintentionally suppress class-discriminative variations that CE alone would preserve. Second, the small dataset magnifies gradient variance. SupCon relies on multiple same-class positive pairs per

**Table 3. Performance comparison of self-supervised, attention-based, and transformer-based models.**

| Model | Backbone | Pretraining | Attention | Loss Function | Test Acc. (%) | TTA Acc. (%) | Macro ROC-AUC | # Params (M) |
|---|---|---|---|---|---|---|---|---|
| Spinach-CNN | Custom CNN | None | None | CE | 91.00 | 96.68 | 0.9921 | 5.49 |
| Spinach-ResSENet | ResNet + SE | None | SE | CE | 96.01 | 95.35 | 0.9963 | 5.53 |
| Spinach-ViT | ViT-B/16 | None | – | CE | 90.7 | 91.3 | 0.9850 | 85.5 |
| SimSiam-ResNet-50 (CE) | ResNet-50 | SimSiam (in-domain) | None | CE | $91.92 \pm 5.34$ | $91.93 \pm 5.35$ | $0.9921 \pm 0.0071$ | 24.56 |
| SimSiam-ResNet-50 (Hybrid) | ResNet-50 | SimSiam (in-domain) | None | CE + $\lambda$·SupCon | $94.61 \pm 1.54$ | $94.62 \pm 1.55$ | $0.9850 \pm 0.0026$ | 24.82 |
| SimSiam-CBAM-ResNet-50 (CE) | ResNet-50 + CBAM | SimSiam (in-domain) | CBAM | CE | $97.31 \pm 1.17$ | $97.33 \pm 1.17$ | $0.9983 \pm 0.0003$ | $27.08 \pm 0.00$ |
| SimSiam-CBAM-ResNet-50 (Hybrid) | ResNet-50 + CBAM | SimSiam (in-domain) | CBAM | CE + $\lambda$·SupCon | $95.29 \pm 0.58$ | $95.30 \pm 0.58$ | $0.9967 \pm 0.0006$ | $27.34 \pm 0.00$ |
| SwinV2-Base (Scratch, Strict Eval) | SwinV2-Base | None (scratch) | Window | CE (label-smoothed) | 94.95 | 95.96 | 0.9970 | 86.90 |
| SwinV2-Base (ms_in22k_ft_in1k, Strict Eval) | SwinV2-Base | ImageNet-22k → ImageNet-1k | Window | CE (label-smoothed) | 97.98 | 98.99 | 1.0000 | 86.90 |

*Note:* **"TTA" stands for Test-Time Augmentation.** "SupCon" refers to Supervised Contrastive Loss. In the hybrid setting, the objective is additive: $\mathcal{L} = \mathcal{L}_{CE} + \lambda \mathcal{L}_{SupCon}$. The TTA policy is selected on the validation set (1/2/4 views) and applied once on the test set; TTA can match single-crop when the selected policy uses one view. "Strict Eval" indicates evaluation via direct resize to 256×256 with no center crop. "–" denotes not reported in this protocol. Parameter count is in millions (M). Reported results are the mean ± standard deviation over 3 random seeds (0,1,2).

batch, but with moderate mini-batch sizes and class-balanced sampling, the number of positives per class can be low and inconsistent. This increases optimization noise and can create gradient conflict between CE and SupCon, especially when the CE-only model is already operating near ceiling performance. Consequently, while the hybrid loss improves representation structure (as reflected in high ROC-AUC), it may act as mild over-regularization during fine-tuning, leading to the observed small decrease in top-1 accuracy.

Finally, the transformer-based models achieved the strongest overall results in this evaluation setting. SwinV2-Base trained from scratch under strict evaluation obtained 94.95% test accuracy and 95.96% TTA accuracy with macro ROC-AUC 0.9970 at 86.90M parameters. The pretrained SwinV2-Base (`ms_in22k_ft_in1k`) achieved the highest performance overall, reaching 97.98% test accuracy and 98.99% TTA accuracy with a perfect macro ROC-AUC of 1.0000, again at 86.90M parameters. Overall, the results show that attention-augmented self-supervised learning (SimSiam-CBAM) provides a strong parameter-efficient alternative to large transformers, while the pretrained Swin model delivers the best absolute accuracy at a substantially higher parameter cost.

## 5 Web deployment

As visualized in Fig 19, the trained SimSiam-CBAM-ResNet-50 model was deployed as a lightweight web application to make it accessible and convenient for farmers to use. The frontend, built with HTML, CSS, and JavaScript, has a simple

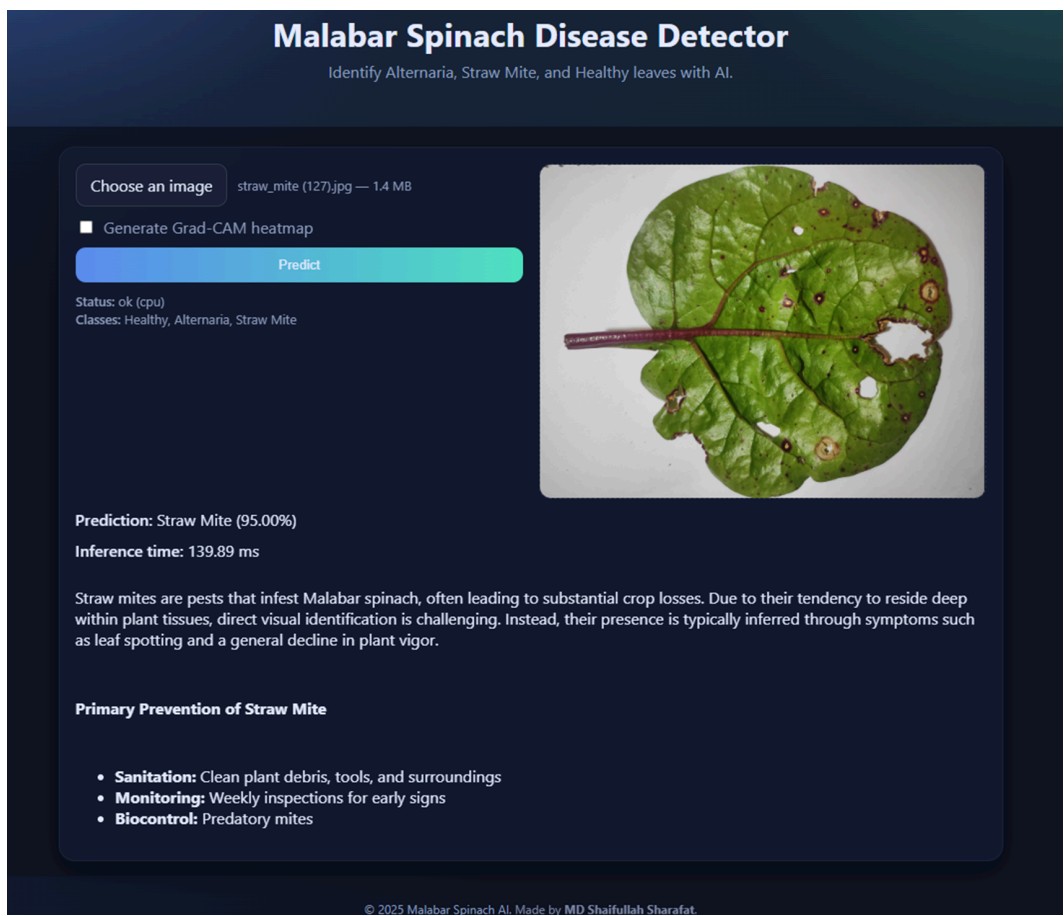

**Fig 19**. Deployment of the lightweight web application for the proposed spinach disease detection.

interface that lets users upload spinach leaf images. After that, users can see the model's prediction in a matter of seconds. The results display the predicted class label (Healthy, Alternaria, or Straw Mite), its associated confidence score, and, upon request, Grad-CAM heatmap visualizations highlighting disease-relevant image regions with an average inference time of under 800 ms.

The backend uses FastAPI, a high-performance Python framework well-suited for low-latency inference. Using the Pillow library, uploaded images are preprocessed, resized to 224×224 pixels, normalized according to ImageNet statistics, and passed through the trained deep learning model. The server responds with a structured JSON payload containing classification probabilities, inference time, and domain-specific annotations (Fig 20).

For Alternaria, Azoxystrobin, Mancozeb, and copper-based compounds are among the fungicides that were recommended by the system. Their FRAC codes and usage notes are also provided [29]. For straw mite infestations, the system suggests preventive actions including sanitation practices, regular field monitoring, and the use of predatory mites as biocontrol agents.

The deployment is completely self-contained with a FastAPI backend that simultaneously serves the static frontend. A `requirements.txt` file lists all dependencies clearly, making the setup easy and reproducible. The web application can run locally on a single machine or be consistently deployed to cloud infrastructure. This way, the system connects academic research with real-world use, providing a real-time digital assistant designed to help manage spinach diseases.

## 6 Model complexity

Beyond accuracy, practical deployment factors such as parameter count, checkpoint size, inference speed, and hardware feasibility play a central role in evaluating agricultural AI systems. All training experiments were conducted using a single NVIDIA Tesla P100 GPU on the Kaggle platform, whereas deployment inference was evaluated on a Ryzen 5 5600G CPU with 16GB DDR4 RAM. Under these conditions, the proposed SimSiam-CBAM-ResNet-50 model achieved an average inference time of under 800ms per image, demonstrating real-time capability without requiring dedicated GPUs or high-end servers.

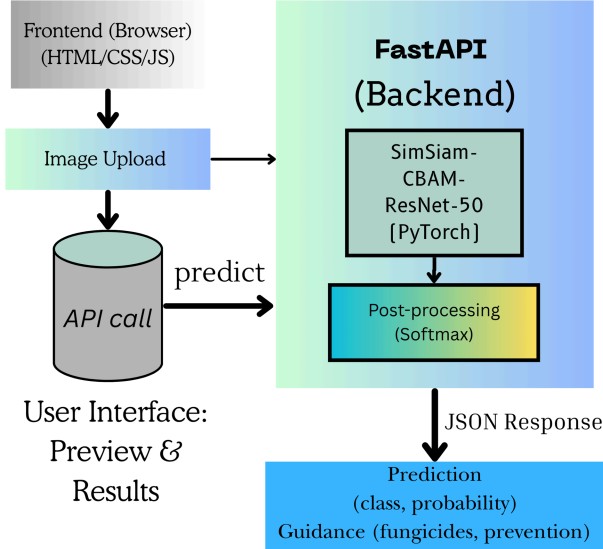

**Fig 20. Web application development architecture.**

**Table 4**. Model complexity and checkpoint size comparison for shortlisted architectures.

| Model | Params (M) | Checkpoint Size (MB) | Test Acc. (%) | Macro ROC-AUC |
|---|---|---|---|---|
| Spinach-ResSENet | 5.53 | 21.13 | 96.01 | 0.9963 |
| SimSiam-ResNet-50 (CE) | 24.56 | 98.57 | 91.92 | 0.9921 |
| SimSiam-ResNet-50 (Hybrid) | 24.82 | 99.62 | 94.61 | 0.9850 |
| SimSiam-CBAM-ResNet-50 (CE) | 27.08 | 108.66 | 97.31 | 0.9983 |
| SimSiam-CBAM-ResNet-50 (Hybrid) | 27.34 | 109.71 | 95.29 | 0.9967 |
| SwinV2-Base (Scratch, Strict Eval) | 86.90 | 195.98 | 94.95 | 0.9970 |
| SwinV2-Base (ImageNet-22k → 1k) | 86.90 | 347.72 | 97.98 | 1.0000 |

To contextualize this performance, Table 4 summarizes model complexity and storage footprint for all shortlisted architectures. These values reflect the final fine-tuned checkpoint sizes, which directly influence deployment feasibility on edge devices, smartphones, and low-power agricultural hardware.

Table 4 highlights a clear accuracy–complexity trade-off. Although the pretrained SwinV2-Base model achieves the highest absolute performance (97.98% test accuracy and 98.99% with TTA), it requires 86.9M parameters and a 347.72MB checkpoint—over **3.2× larger** than the proposed SimSiam-CBAM-ResNet-50. In contrast, SimSiam-CBAM-ResNet-50 maintains a compact footprint (27.08M parameters, 108.66MB checkpoint) while delivering 97.31% accuracy and a near-saturated macro ROC-AUC of 0.9983. The model therefore achieves one of the best balances between accuracy, robustness, interpretability, and computational efficiency, making it the most practical choice for real-world deployment in resource-limited agricultural environments.

## 7 Limitations

Despite achieving strong performance, this study has several limitations. First, the dataset consists exclusively of Malabar spinach leaf images collected within Bangladesh, which may introduce geographic or environmental bias; cross-regional generalization has not yet been evaluated. Although the proposed model was deployed in a lightweight FastAPI web application and demonstrated sub-800 ms inference on a Ryzen 5 5600G CPU, no usability or field-level studies have yet been conducted with farmers. Thus, the real-world adoption, decision-making impact, and reliability under varied farming conditions remain untested. Additionally, further dataset expansion, both in class diversity and sample size, would make it more robust and lower the risk of overfitting. Finally, due to the absence of prior deep learning work on Malabar spinach disease detection, direct performance benchmarking against existing methods was not possible.

Although the dataset was carefully curated and all disease labels were verified by agricultural experts from Habiganj Agricultural University and Daffodil International University, the images originate exclusively from Bangladesh. As a result, the dataset reflects local environmental conditions such as lighting, humidity, leaf appearance, background clutter, and cultivar-specific morphology. While the agronomic symptoms of Alternaria and straw mite are generally consistent across regions, geographic and climatic factors may influence the visual presentation of disease, which could affect generalization to other global contexts. Future work will address this by incorporating cross-regional samples, exploring domain-invariant feature learning, and evaluating the model on datasets collected under different climatic and agricultural settings.

## 8 Conclusion and future work

This study presented an integrated deep learning framework for automated detection of Alternaria and straw mite diseases in Malabar spinach leaves. Pretrained convolutional models such as EfficientNetB0 and ResNet-50 achieved strong performance (above 95% accuracy) but remain relatively large and resource-intensive. Our custom architectures, including the CNN-based SpinachCNN and Spinach-ResSENet with squeeze-and-excitation blocks, provided competitive accuracy with substantially fewer parameters. Although the Swin Transformer attained the best overall performance with

a test accuracy of 97.98% and a TTA accuracy of 98.99%, its reliance on ImageNet-22k pretraining and its large parameter count of 86.9 million make it less suitable for many deployment-constrained settings. In comparison, the proposed SimSiam-CBAM-ResNet-50, pretrained in a self-supervised manner on unlabeled Malabar spinach images, achieved a competitive 97.31% test accuracy and a macro ROC-AUC of 0.9983. Its channel-spatial attention mechanism enhances interpretability, while the self-supervised pretraining and CE-based fine-tuning support strong generalization. Ablation studies under Gaussian and salt-and-pepper noise further confirmed its robustness, making it a practical, lightweight, and domain-optimized option for real-world farming applications, particularly on edge and mobile devices.

Future work will focus on adapting to different spinach varieties, climates, and regions using domain-invariant feature learning and style transfer. Subsequent research will focus on the lightweight deployment of the SimSiam-CBAM-ResNet-50 model on edge computing platforms, i.e., smartphones, Raspberry Pi, NVIDIA Jetson, etc. We will use optimization strategies such as model pruning, quantization, and TensorRT acceleration to support real-time inference. Additionally, we will explore using multimodal data, particularly spectral and thermal imagery, to allow early detection of disease symptoms beyond the visible spectrum. This approach aims to improve diagnostic accuracy and support timely, data-driven interventions in field conditions. Future studies will also include field trials to evaluate usability, user experience, and decision-making support of the proposed leaf disease classification system for farmers in real-world settings.

## Author contributions

**Conceptualization:** Nilavro Das Kabya, MD Shaifullah Sharafat, Riasat Khan.

**Data curation:** Nilavro Das Kabya, MD Shaifullah Sharafat, Rahimul Islam Emu, Riasat Khan.

**Formal analysis:** Nilavro Das Kabya, MD Shaifullah Sharafat, Rahimul Islam Emu.

**Funding acquisition:** Riasat Khan.

**Investigation:** Nilavro Das Kabya, MD Shaifullah Sharafat, Rahimul Islam Emu, Mehrab Karim Opee.

**Methodology:** Nilavro Das Kabya, MD Shaifullah Sharafat, Rahimul Islam Emu, Mehrab Karim Opee.

**Project administration:** Riasat Khan.

**Software:** Nilavro Das Kabya, MD Shaifullah Sharafat, Rahimul Islam Emu.

**Supervision:** Riasat Khan.

**Validation:** Nilavro Das Kabya, MD Shaifullah Sharafat, Rahimul Islam Emu, Mehrab Karim Opee.

**Visualization:** Nilavro Das Kabya, MD Shaifullah Sharafat, Mehrab Karim Opee, Riasat Khan.

**Writing – original draft:** Nilavro Das Kabya, MD Shaifullah Sharafat, Rahimul Islam Emu, Mehrab Karim Opee.

**Writing – review & editing:** Riasat Khan.

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
