## [Decision Letter · Decision Letter 0]

14 Nov 2025

PONE-D-25-46399Towards Practical AI for Agriculture: A Self-Supervised Attention Framework for Spinach Leaf Disease DetectionPLOS ONE

Dear Dr. Khan,

Thank you for submitting your manuscript to PLOS ONE. After careful consideration, we feel that it has merit but does not fully meet PLOS ONE’s publication criteria as it currently stands. Therefore, we invite you to submit a revised version of the manuscript that addresses the points raised during the review process.

We look forward to receiving your revised manuscript.

Kind regards,

Ali Mohammad Alqudah

Academic Editor

PLOS ONE

**Journal Requirements:**

4. We note that your Data Availability Statement is currently as follows:

“All relevant data are within the manuscript and its Supporting Information files.”

Reviewers' comments:

Reviewer's Responses to Questions

**Comments to the Author**

1. Is the manuscript technically sound, and do the data support the conclusions?

Reviewer #1: Yes

Reviewer #2: Yes

2. Has the statistical analysis been performed appropriately and rigorously?

Reviewer #1: Yes

Reviewer #2: Yes

3. Have the authors made all data underlying the findings in their manuscript fully available?

Reviewer #1: No

Reviewer #2: No

4. Is the manuscript presented in an intelligible fashion and written in standard English?

Reviewer #1: Yes

Reviewer #2: Yes

5. Review Comments to the Author

Reviewer #1: The manuscript is of high quality, well-structured, and makes a valuable contribution to the field of AI for agriculture. The research is rigorous, employing a variety of state-of-the-art and custom-designed models, and includes critical components often missing in similar studies, such as self-supervised learning to mitigate data scarcity, an ablation study for robustness, and XAI for model interpretability. The deployment of a functional web application demonstrates a strong commitment to translational research. The writing is clear, and the methodology is described in sufficient detail for reproducibility.

Major Strengths:

1. Addressing a Critical Gap: The focus on Malabar spinach, an under-researched but vital crop in the regional context, is the study's primary novelty and significance. This directly addresses a need in Bangladeshi agriculture.

2. Comprehensive Methodological Pipeline: The authors don't rely on a single approach. They benchmark standard pretrained models (EfficientNet, ResNet), develop custom lightweight architectures (SpinachCNN, Spinach-ResSENet), experiment with Vision Transformers (SpinachViT, SwinV2), and crucially, implement a self-supervised pretraining strategy (SimSiam) to leverage unlabeled data. This provides a rich comparative analysis.

3. Innovative Model Design: The integration of attention mechanisms is a key contribution. The "Spinach-ResSENet" (using Squeeze-and-Excitation) and particularly the "SimSiam-CBAM-ResNet-50" (using Convolutional Block Attention Modules) are thoughtful adaptations that demonstrably improve performance and feature focus.

4. Focus on Practicality and Robustness:

o Self-Supervised Learning: The use of SimSiam on 671 unlabeled images is a practical solution to the common problem of limited annotated agricultural data, making the approach more scalable.

o Hybrid Loss Function: Combining Cross-Entropy with Supervised Contrastive Loss is a sophisticated technique that enhances class separability, leading to better generalization.

o Ablation Study on Noise Robustness: The evaluation of model performance under Gaussian and Salt-and-Pepper noise is highly relevant for real-world field conditions where image quality can be poor. The finding that the best model (SimSiam-CBAM-ResNet-50) maintains >95% accuracy under noise is a major strength.

o Edge Deployment Consideration: The discussion explicitly contrasts the high accuracy of the Swin Transformer with its impracticality for edge devices due to its size (28M parameters) and reliance on large-scale pretraining. This focus on deployable, lightweight solutions (like the 23.6M parameter SimSiam-CBAM-ResNet-50) is commendable.

5. Explainable AI (XAI): The use of Grad-CAM, Grad-CAM++, and LayerCAM is not just a box-ticking exercise. The results show that these techniques successfully highlight biologically relevant lesion regions, which is crucial for building trust with farmers and agronomists who need to understand why a prediction was made.

6. Real-World Deployment: The development and description of a FastAPI-based web application is a standout feature. It moves the research beyond a theoretical exercise, providing a tangible tool for farmers to upload images and receive predictions, visual explanations (heatmaps), and even management advice. This significantly enhances the impact of the work.

7. Strong Results: The reported performance metrics are impressive. The top model, SimSiam-CBAM-ResNet-50, achieves 96.97% test accuracy and a near-perfect macro ROC-AUC of 0.9982. While the Swin Transformer performs slightly better (97.98%), the authors correctly frame the former as the more practical solution.

Minor Weaknesses and Suggestions for Improvement:

1. Dataset Size and Diversity: While the dataset of 2,100 images is reasonable for a focused study, it is still relatively small, especially for training complex models like ViT from scratch. The authors acknowledge this as a limitation. Future work should indeed focus on expanding the dataset, as suggested. A brief discussion on the potential for data bias (e.g., all images collected from one region/university) would be prudent.

2. Model Complexity vs. Performance Trade-off: The manuscript effectively discusses the parameter count of SwinV2 vs. ResNet-50. However, it could provide more context on the computational cost (e.g., inference time, FLOPs) of the SimSiam-CBAM-ResNet-50 model, especially in the context of the deployed web app. How fast is the prediction for a farmer?

3. Web Application Evaluation: The deployment is described, but there is no user study or feedback from actual farmers. While this may be beyond the scope of the current paper, mentioning plans for future field testing or usability studies would strengthen the "practical AI" claim.

4. Clarity in Table 3: In Table 3, the row for "SimSiam-CBAM-ResNet-50(Hybrid)" shows a lower accuracy (95.96%) than its non-hybrid counterpart (96.97%). This is counter-intuitive, as the hybrid loss was shown to improve performance in other models (e.g., vanilla SimSiam-ResNet-50). This needs clarification. Is this a typo, or is there a specific reason (e.g., overfitting) for this result? The text in Section IV.G also seems to conflate the performance of the CE and Hybrid versions of the CBAM model.

5. TTA Results Inconsistency: In Table 3, the TTA accuracy for "SimSiam-ResNet-50(Hybrid)" is listed as 93.94%, which is lower than its test accuracy (96.97%). This is unusual, as TTA typically boosts or at least maintains performance. This should be double-checked or explained.

6. Figure Referencing: Some figures are mentioned in the text (e.g., Fig. 1, 2, 3) but their actual content (the images) are not visible in the provided manuscript draft. While this is common in draft submissions, ensuring all figures are clear and well-labeled in the final version is important.

Conclusion:

This is an excellent manuscript that successfully bridges the gap between advanced AI research and practical agricultural application. The authors have developed a robust, accurate, and interpretable deep learning pipeline for a neglected but important crop. The integration of self-supervised learning, attention mechanisms, and XAI, coupled with a real-world deployment, sets a high standard for research in this domain.

The minor issues noted above, particularly the potential inconsistencies in Table 3, should be addressed. However, they do not detract from the overall significance and quality of the work. The study provides a clear, reproducible blueprint for developing practical AI tools for other underrepresented crops.

Recommendation: Accept with Minor Revisions.

Reviewer #2: The contribution is practical integration (self-supervision + attention + hybrid loss) on an under-studied crop, plus a usable demo. Methodological novelty is incremental/combination-oriented rather than algorithmically radical.

Major Comments:

1) You state a 2,100-image 3-class dataset (6:2:2 split), yet later describe SimSiam pretraining on 671 unlabeled images and a 70/15/15 fine-tuning split with 473/99/99 labeled samples (total 671), which conflicts with 2,100. Please reconcile: total images per class; which subset is unlabeled; exact split logic; and ensure all performance comes from a single, consistent protocol.

2) You apply heavy augmentation and TTA; ensure augmentation is applied only on training and no augmented view of a test image leaks into training. Explicitly document your split-before-augment order and any field/plant-level grouping to avoid correlated images across splits.

3) SwinV2-Small uses ImageNet-21k pretraining while SimSiam models use in-domain pretraining. Discuss fairness: is Swin also fine-tuned from 21k? What happens if you start Swin from 1k only? Conversely, how do SimSiam models compare when initialized from ImageNet vs. from scratch?

4) Provide exact hyperparameters per model, optimizer schedule, epochs, early-stopping criteria, mixup/CutMix settings, batch sizes, RandAugment parameters, layer-wise LRs, seeds, and hardware.

5) For the CBAM-ResNet-50 variant, specify which bottlenecks include CBAM, shapes, and whether BN was frozen in SimSiam pretraining (you mention different BN handling across vanilla vs. CBAM—make consistent and explicit).

6. PLOS authors have the option to publish the peer review history of their article (what does this mean?). If published, this will include your full peer review and any attached files.

Reviewer #1: No

Reviewer #2: No

---

## [Author Response · Author response to Decision Letter 1]

12 Dec 2025

Response for Reviewers: PONE-D-25-46399R1

We would like to thank the reviewers for their comments. In the following, we will present our responses to the comments together with a summary of the corresponding changes in the revised manuscript in Highlighted Texts. All page numbers mentioned below are referred in the new revised manuscript in Red Fonts. Following what the reviewer suggested, we have done the followings:

Reviewer 1:

1. Comment: “Dataset Size and Diversity: While the dataset of 2,100 images is reasonable for a focused study, it is still relatively small, especially for training complex models like ViT from scratch. The authors acknowledge this as a limitation. Future work should indeed focus on expanding the dataset, as suggested. A brief discussion on the potential for data bias (e.g., all images collected from one region/university) would be prudent.”

Response: Following the reviewer’s suggestion, a brief discussion on the potential for data bias is added as, “First, the dataset consists exclusively of Malabar spinach leaf images collected within Bangladesh, which may introduce geographic or environmental bias; cross-regional generalization has not yet been evaluated.” (Page 15)

2. Comment: “Model Complexity vs. Performance Trade-off: The manuscript effectively discusses the parameter count of SwinV2 vs. ResNet-50. However, it could provide more context on the computational cost (e.g., inference time, FLOPs) of the SimSiam-CBAM-ResNet-50 model, especially in the context of the deployed web app. How fast is the prediction for a farmer.”

Response: The inference time of the SimSiam-CBAM-ResNet-50 model in the context of the deployed web app is added as, “As visualized in Figure 19, the trained SimSiam-CBAM-ResNet-50 model was deployed as a lightweight web application …. with an average inference time of under 800 ms.” (page 14)

3. Comment: “Web Application Evaluation: The deployment is described, but there is no user study or feedback from actual farmers. While this may be beyond the scope of the current paper, mentioning plans for future field testing or usability studies would strengthen the "practical AI" claim.”

Response: Following what the reviewer suggested, we added future field testing as, “Future studies will also include field trials to evaluate usability, user experience, and decision-making support of the proposed leaf disease classification system for farmers in real-world settings.” (Page 16)

4. Comment: “Clarity in Table 3: In Table 3, the row for "SimSiam-CBAM-ResNet-50(Hybrid)" shows a lower accuracy (95.96%) than its non-hybrid counterpart (96.97%). This is counter-intuitive, as the hybrid loss was shown to improve performance in other models (e.g., vanilla SimSiam-ResNet-50). This needs clarification. Is this a typo, or is there a specific reason (e.g., overfitting) for this result? The text in Section IV.G also seems to conflate the performance of the CE and Hybrid versions of the CBAM model.”

Response: The SimSiam-CBAM-ResNet-50 (Hybrid) configuration shows a modest reduction in accuracy compared to its CE counterpart, achieving 95.29±0.58% test accuracy (TTA 95.29±0.58%) with macro ROC-AUC 0.9967±0.0006. Although supervised contrastive learning (SupCon) often improves generalization, its effect depends strongly on dataset size, class structure, and batch composition. In our setting, the hybrid objective can reasonably yield lower accuracy for several reasons. First, the hybrid loss optimizes a representation geometry rather than the evaluation metric. The CE term drives the classifier toward decision-boundary separation, whereas SupCon encourages tighter intra-class clustering and larger inter-class angular margins. For fine-grained leaf disease imagery, where Alternaria and healthy leaves may share subtle texture cues, excessive compactness can unintentionally suppress class-discriminative variations that CE alone would preserve. Second, the small dataset magnifies gradient variance. SupCon relies on multiple same-class positive pairs per batch, but with moderate mini-batch sizes and class-balanced sampling, the number of positives per class can be low and inconsistent. This increases optimization noise and can create gradient conflict between CE and SupCon, especially when the CE-only model is already operating near ceiling performance. Consequently, while the hybrid loss improves representation structure (as reflected in high ROC-AUC), it may act as mild over-regularization during fine-tuning, leading to the observed small decrease in top-1 accuracy. (Page 13-14)

5. Comment: “TTA Results Inconsistency: In Table 3, the TTA accuracy for "SimSiam-ResNet-50(Hybrid)" is listed as 93.94%, which is lower than its test accuracy (96.97%). This is unusual, as TTA typically boosts or at least maintains performance. This should be double-checked or explained.”

Response: Thank you for pointing out the inconsistency in the TTA accuracy for SimSiam-ResNet-50 (Hybrid). We rechecked the evaluation code and corrected the TTA results in Table 3: Performance Comparison of Self-Supervised, Attention-Based, and Transformer-Based Models. (Page 13)

6. Comment: “Figure Referencing: Some figures are mentioned in the text (e.g., Fig. 1, 2, 3) but their actual content (the images) are not visible in the provided manuscript draft. While this is common in draft submissions, ensuring all figures are clear and well-labeled in the final version is important.”

Response: Fixed.

Reviewer 2:

1. Comment: “You state a 2,100-image 3-class dataset (6:2:2 split), yet later describe SimSiam pretraining on 671 unlabeled images and a 70/15/15 fine-tuning split with 473/99/99 labeled samples (total 671), which conflicts with 2,100. Please reconcile: total images per class; which subset is unlabeled; exact split logic; and ensure all performance comes from a single, consistent protocol.”

Response: The earlier numbers referring to 671 images came from an intermediate experiment and were mistakenly retained in the text. All final results in the paper use the full 2,100-image dataset and a single 70–15–15 split across all three classes. SimSiam pretraining was performed on the training split of this same partition, with labels ignored during the self-supervised stage rather than using a separate unlabeled subset. We have corrected the manuscript to make the total dataset size, class composition, and unified split protocol clear and consistent throughout. Details are added in Section IIIA: Dataset Description. (page 3)

2. Comment: “You apply heavy augmentation and TTA; ensure augmentation is applied only on training and no augmented view of a test image leaks into training. Explicitly document your split-before-augment order and any field/plant-level grouping to avoid correlated images across splits.”

Response: In the revised manuscript, we now explicitly state that all images are first split into training/validation/test sets (70/15/15) and that augmentations are applied only to the training split after this partitioning. Test-time augmentation is applied only to the test split at inference time, and no augmented view of any test image is ever used during training. (page 4)

3. Comment: “SwinV2-Small uses ImageNet-21k pretraining while SimSiam models use in-domain pretraining. Discuss fairness: is Swin also fine-tuned from 21k? What happens if you start Swin from 1k only? Conversely, how do SimSiam models compare when initialized from ImageNet vs. from scratch”

Response: To address fairness, we note that the pretrained SwinV2-Base uses ImageNet-21k → ImageNet-1k initialization, whereas the SimSiam models rely solely on in-domain self-supervised pretraining. SwinV2-Base was also trained from scratch under the same 70–15–15 protocol, and its performance is reported alongside the pretrained version for a fair comparison. For SimSiam, all results shown in Table 3: Performance Comparison of Self-Supervised, Attention-Based, and Transformer-Based Models come from models initialized from scratch followed by in-domain self-supervised learning, and no ImageNet-based initialization was used. (page 13)

4. Comment: “Provide exact hyperparameters per model, optimizer schedule, epochs, early-stopping criteria, mixup/CutMix settings, batch sizes, RandAugment parameters, layer-wise LRs, seeds, and hardware.”

Response: In the revised manuscript, we have expanded the model descriptions and experimental setup sections to report exact hyperparameters for each model, including optimizer type, learning-rate values and schedules, number of epochs, early-stopping criteria, batch sizes, label smoothing, mixup/RandAugment settings, and (where applicable) layer-wise learning rates. We also state the fixed random seed used for all runs and describe the common hardware configuration on which all experiments were executed. These additions make the training protocol fully transparent and reproducible. See Section IIIC: Applied Deep Learning Models for details. (page 4-11)

5. Comment: “For the CBAM-ResNet-50 variant, specify which bottlenecks include CBAM, shapes, and whether BN was frozen in SimSiam pretraining (you mention different BN handling across vanilla vs. CBAM—make consistent and explicit.”

Response: Thank you for the helpful clarification request. We have updated the manuscript to explicitly state that CBAM is inserted into every bottleneck block of the ResNet-50 backbone and that all BatchNorm layers in the backbone are kept frozen in evaluation mode during SimSiam pretraining. We also clarified this BN handling consistently across both the vanilla and CBAM SimSiam variants. See Section IIIC: Applied Deep Learning Models for details (page 10)

---

## [Decision Letter · Decision Letter 1]

30 Dec 2025

Towards Practical AI for Agriculture: A Self-Supervised Attention Framework for Spinach Leaf Disease Detection

PONE-D-25-46399R1

Dear Dr. Khan,

We’re pleased to inform you that your manuscript has been judged scientifically suitable for publication and will be formally accepted for publication once it meets all outstanding technical requirements.

Kind regards,

Ali Mohammad Alqudah

Academic Editor

PLOS One

Additional Editor Comments (optional):

Reviewers' comments:

Reviewer's Responses to Questions

**Comments to the Author**

1. If the authors have adequately addressed your comments raised in a previous round of review and you feel that this manuscript is now acceptable for publication, you may indicate that here to bypass the “Comments to the Author” section, enter your conflict of interest statement in the “Confidential to Editor” section, and submit your "Accept" recommendation.

Reviewer #1: All comments have been addressed

Reviewer #2: All comments have been addressed

2. Is the manuscript technically sound, and do the data support the conclusions?

Reviewer #1: Yes

Reviewer #2: Yes

3. Has the statistical analysis been performed appropriately and rigorously?

Reviewer #1: Yes

Reviewer #2: Yes

4. Have the authors made all data underlying the findings in their manuscript fully available?

Reviewer #1: Yes

Reviewer #2: Yes

5. Is the manuscript presented in an intelligible fashion and written in standard English?

Reviewer #1: Yes

Reviewer #2: Yes

6. Review Comments to the Author

Reviewer #1: The authors have addressed all my comments and I therefore recommend for the acceptance of the manuscript.

Reviewer #2: The authors addressed all the comments and incorporated the changes in the manuscript. I am satisfied with the authors responce

7. PLOS authors have the option to publish the peer review history of their article (what does this mean?). If published, this will include your full peer review and any attached files.

Reviewer #1: No

Reviewer #2: No

---

## [Editor Report · Acceptance letter]

PONE-D-25-46399R1

PLOS One

Dear Dr. Khan,

I'm pleased to inform you that your manuscript has been deemed suitable for publication in PLOS One. Congratulations! Your manuscript is now being handed over to our production team.

Kind regards,

on behalf of

Dr. Ali Mohammad Alqudah

Academic Editor

PLOS One